# Angptl8 mediates food-driven resetting of hepatic circadian clock in mice

Siyu Chen[1,2,4], Mengyang Feng[1,4], Shiyao Zhang[2], Zhewen Dong[2], Yifan Wang[2], Wenxiang Zhang[1,2] & Chang Liu [1,2,3]

Diurnal light-dark cycle resets the master clock, while timed food intake is another potent synchronizer of peripheral clocks in mammals. As the largest metabolic organ, the liver sensitively responds to the food signals and secretes hepatokines, leading to the robust regulation of metabolic and clock processes. However, it remains unknown which hepatokine mediates the food-driven resetting of the liver clock independent of the master clock. Here, we identify Angptl8 as a hepatokine that resets diurnal rhythms of hepatic clock and metabolic genes in mice. Mechanistically, the resetting function of Angptl8 is dependent on the signal relay of the membrane receptor PirB, phosphorylation of kinases and transcriptional factors, and consequently transient activation of the central clock gene *Per1*. Importantly, inhibition of Angptl8 signaling partially blocks food-entrained resetting of liver clock in mice. We have thus identified Angptl8 as a key regulator of the liver clock in response to food.

[1] State Key Laboratory of Natural Medicines, China Pharmaceutical University, Nanjing, 211198 Jiangsu, China. [2] School of Life Science and Technology, China Pharmaceutical University, Nanjing, 211198 Jiangsu, China. [3] State key Laboratory of Pharmaceutical Biotechnology, Nanjing University, Nanjing, 211198 Jiangsu, China. [4]These authors contributed equally: Siyu Chen, Mengyang Feng. Correspondence and requests for materials should be addressed to C.L. (email: changliu@cpu.edu.cn)

Mammalian circadian clocks precisely control the rhythms of behavior and physiology, and can be reset by various environmental signals. While the light–dark (LD) cycle resets the master clock, timed food intake is a potent synchronizer of peripheral clocks[1]. In mammals, the circadian clock system comprises ubiquitously expressed cellular inter-connecting transcriptional–translational loops of clock genes/proteins[2]. In the core loop, the transcriptional activators of circadian locomotor output cycles kaput (CLOCK) and brain and muscle ARNT-like 1 (BMAL1) form heterodimers that activate the expression of repressors, encoded by three Period (Per 1–3) and two Cryptochrome (Cry 1/2) genes. Towards the end of the day, PER and CRY protein complexes translocate into the nucleus and inhibit CLOCK/BMAL1-mediated transcription. Due to their instability, the protein and mRNA abundance of these repressors rapidly decreases below the threshold required for autorepression, clearing the way for a new cycle[3–5]. In addition, the orphan nuclear receptors of the retinoic acid receptor-related orphan receptor (ROR) and REV-ERB families are also implicated in the control of circadian clock function[6].

Epidemiological investigations have indicated that prolonged light exposure or rotating shift work is closely correlated to an elevated risk for developing various diseases, including cancer, cardiovascular disease, immune deficiency, and metabolic disorders[7]. These observations suggest that altered sleep/wake schedules may disrupt endogenous circadian clock systems through the mistimed resetting of certain time givers, the so called Zeitgebers. The external LD cycle is the most dominant Zeitgeber to reset the central circadian pacemaker located in the hypothalamic suprachiasmatic nucleus (SCN). The SCN receives light information from the retina and generates central clock signals upon light stimulation to drive slave oscillators distributed in various peripheral tissues through behavioral and neuroendocrine transmitters[1,8]. Unsurprisingly, the SCN itself is only responsive to photic timing signals[8]. However, feeding rhythms and nutritional signals play a key role in the regulation of peripheral clock function. Time-restricted feeding (e.g. food access is limited within the normal rest phase of an organism) uncouples peripheral clocks from the SCN for adaptation to the new timing of food availability[9]. This uncoupling might importantly contribute importantly to the pathogenesis of metabolic diseases found in people with abnormal life routine styles[10]. Animal studies also demonstrated that food intake during the normal rest phase promotes obesity in rodents[11]. However, the mechanisms of food-dependent peripheral clock resetting has not been thoroughly elucidated to date. Specifically, as the largest metabolic organ, the liver sensitively responds to the food signals and secrets hepatokines, leading to the robust regulation of metabolic and clock processes[12–14]. For example, fibroblast growth factor (FGF) 21 is a liver-secreted, insulin-responsive factor that regulates glucose and lipid metabolism[15]. The overexpression of FGF21 in mice increases systemic glucocorticoid levels, suppresses physical activity and alters circadian behavior. However, mice lacking the gene encoding β-Klotho (Klb) in the SCN are refractory to these effects, suggesting that the diverse physiologic and pharmacologic actions of FGF21 are still mediated by the nervous system[16–18]. In contrast, the entrainment of peripheral clocks is independent on the SCN, and complex foods show much stronger effects on the phase resetting compared to carbohydrate intake, indicating that other factors must be involved[19]. These remaining questions prompt us to identify hepatokines that mediate the food-driven resetting of the liver clock, independent of the master clock.

To answer these questions, we cluster high-throughput RNA sequencing results from liver samples collected in mice subjected to an overnight fasting or constant darkness (DD) and find that the hepatokine Angptl8 shows daily fluctuation in both the liver and serum of mice. Angptl8 (also known as betatrophin, Lipasin, and C19orf80) belongs to the angiopoietin-like protein family, which includes 8 proteins structurally similar to the angiopoietins and are involved in the regulation of lipid metabolism, inflammation, cancer cell invasion, and hematopoietic stem activity[20–23]. While Angptl8 is highly enriched in the liver and adipocytes of mice, this protein is exclusively expressed in the liver of humans[24,25]. Angptl8 has been shown to play critical roles in the development of various metabolic diseases. For instance, the serum levels of Angptl8 protein are increased in type 1/2 diabetic patients, and are also positively correlated with total cholesterol (TC), LDL cholesterol and apolipoprotein B levels in morbidly obese subjects and individuals with type 2 diabetes[26,27]. Upon administration of a high-fat diet, Angptl8 mRNA is induced in the liver and brown and white adipose tissues of mice. Consistently, Angptl8 overexpression by adenoviruses in mice increases serum triglyceride (TG) levels[24]. In addition to the metabolic functions of Angptl8, a recent study reports that this factor is rhythmically expressed in the mouse liver under the control of glucocorticoid signaling and LXRα pathway, raising the possibility that Angptl8 is involved in the regulation of diurnal oscillation of metabolic processes[28].

In the present study, we find that intraperitoneal (i.p.) injection of recombinant Angptl8 in mice alters diurnal rhythms of locomotor activity, as well as hepatic clock and metabolic genes. In addition, Angptl8-shock induces the rhythmic expression of clock genes in mouse Hepa1c1c-7 hepatoma cells. Mechanistically, the resetting function of Angptl8 is dependent on the signal relay of the membrane receptor PirB, phosphorylation of kinases and transcriptional factors, and consequently transient activation of the central clock gene Per1. On the other hand, as the liver is the major site for Angptl8 expression and secretion, we construct mice with liver-specific Angptl8 knockdown by using adeno-associated virus serotype 8 (AAV8) system harboring an Angptl8 shRNA. We find that liver-specific Angptl8 knockdown partially attenuates food-entrained resetting of liver clock in mice. Similar results are observed when circulating Angptl8 is quenched by Angptl8-neutralizd antibody. Thus, Angptl8 is a key regulator of the liver clock in response to food.

## Results

**Angptl8 responses to food signals and peripheral clocks**. To address the question mentioned above, we performed transcriptional profiling in livers dissected from mice subjected to fasting/refeeding cycles or DD for 6 weeks by using high-throughput RNA sequencing. These analyses revealed a cluster of six rhythmically expressed genes that were regulated by food signals (Fig. 1a, b, Supplementary Data 1–3, Supplementary Table 1–4). Moreover, all of six genes could be reversed by restricted feeding in the mouse liver (Fig. 1c and Supplementary Fig. 1a–f). In particular, Angptl8 is a unique protein that contains an extra-cellular region and can be secreted into circulation as a hepatokine[24,29]. This protein has also attracted much attention in recent years due to its diverse functions in the regulation of lipid and glucose homeostasis[30]. Consistently, the oscillation of serum Angptl8 levels was also reversed by restricted feeding (Fig. 1d). Thus, our research target has been narrowed to and focused on Angptl8. Next, independent RT-qPCR analysis confirmed the expression patterns of hepatic Angptl8 during DD and starvation (Supplementary Fig. 1g and 1h). Specifically, the mRNA expression of Angptl8 in the mouse liver peaked at circadian time (CT) 9 and declined thereafter, showing the lowest expression at CT17. However, hepatic Angptl8 mRNA expression decreased upon

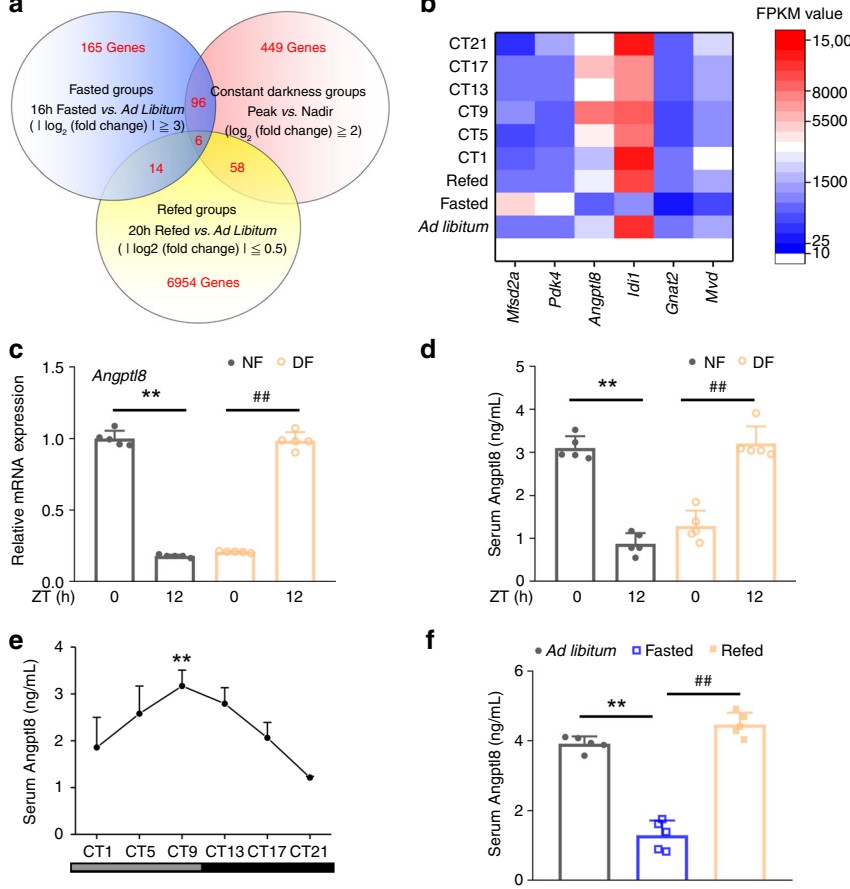

**Fig. 1** Angptl8 responses to food signals and peripheral clocks. **a** Venn diagrams of high-throughput RNA sequencing results from liver samples of mice subjected to 16-h fasting, 16-h fasting followed by 20-h refeeding, or DD. **b** Heat map of six clustered genes derived from high-throughput RNA sequencing results. **c** qPCR analyses of Angptl8 mRNA expression levels in the liver of mice subjected to time-restricted feeding. **d** Serum Angptl8 levels. For **c**, **d** $n = 5$, **$P < 0.01$ NF12 vs. NF0, ##$P < 0.01$ DF12 vs. DF0, one-way ANOVA followed by Bonferroni's post hoc test. **e** Serum Angptl8 levels in mice subjected to DD. $n = 5$, **$P < 0.01$, Peak vs. Nadir, one-way ANOVA followed by Bonferroni's post hoc test. **f** Serum Angptl8 levels in mice subjected to 16-h fasting or 16-h fasting, followed by 20-h refeeding. $n = 5$, **$P < 0.01$ 16-h fasted vs. ad libitum, ##$P < 0.01$ 16-h fasted vs. refed, one-way ANOVA followed by Bonferroni's post hoc test. All data are presented as the means ± SD. Source data are provided as a Source Data file

fasting and recovered after refeeding. The serum levels of Angptl8 showed a similar trend (Fig. 1e, f).

**Angptl8 resets liver clock in vivo.** To evaluate the potential roles of Angptl8 in the regulation of the liver clock, we i.p. injected recombinant Angptl8 into mice for 15 consecutive days at ZT2 (ZT2 is the time point when the mice enter the rest phase, and the serum levels of Angptl8 are low), and collected serum samples at ZT13. While Angptl8 administration at different doses did not alter the mouse body weight and food intake (Supplementary Fig. 2a and 2b), this treatment indeed increased the serum levels of Angptl8 in a dose-dependent manner (Supplementary Fig. 2c). In addition, Angptl8 injection had a modest effect on TG and TC serum levels, as well as blood glucose levels (Supplementary Fig. 2d–2f). We selected 1 μg/kg of Angptl8 for the following in vivo experiments because exogenous Angptl8 injection at this dose best mimicked the peak levels of Angptl8 physiological oscillation in the serum. Interestingly, analysis of mouse wheel-running activities revealed that Angptl8 injection shortened the circadian period ($23.68 \pm 0.09$ h vs. $23.92 \pm 0.12$ h, $P = 0.004$) and advanced the phase of activity rhythms (1.67 h advance, Fig. 2a–c). At the molecular level, while Angptl8 administration did not alter the amplitude of clock gene oscillation in the mouse liver, it led to a significant phase advance (Fig. 2d, detailed phase

and amplitude assessments were presented in Supplementary Table 5). Similar trends were observed in the expression rhythmicity of metabolic genes (Fig. 2e). Contrasting, the rhythmicity of clock gene mRNA expression in the SCN remained unchanged (Supplementary Fig. 3a). In addition, Angptl8 injection had a modest effect on the diurnal rhythms of serum TG, TC and glucose levels (Fig. 2f).

**Angptl8 resets liver clock in vitro.** Certain metabolites, such as glucose and glucocorticoids, induce the circadian expression of key clock genes in cultured cells by a flash exposure (e.g., 2 h)[31,32]. This procedure, referred to as shock, is universally used to examine the clock resetting function of these metabolites in vitro[33]. Similarly, we treated mouse Per2::Luc U2OS cells with recombinant Angptl8 for 2 h. Real-time bioluminescence analyses revealed that Angptl8-shock induced a 23.95-h circadian period of Per2::Luc in these cells (Fig. 3a–c). This transient shock also led to a circadian oscillation of the clock genes in mouse Hepa1c1c-7 cells (Fig. 3d and Supplementary Table 6), and the effects of Angptl8 on evoking clock oscillation were surprisingly comparable to those of 50% horse serum shock, a classic clock synchronizer. In contrast, neither the negative control PBS nor the positive control BSA could induce the expression rhythmicity of clock genes (Supplementary Fig. 3b).

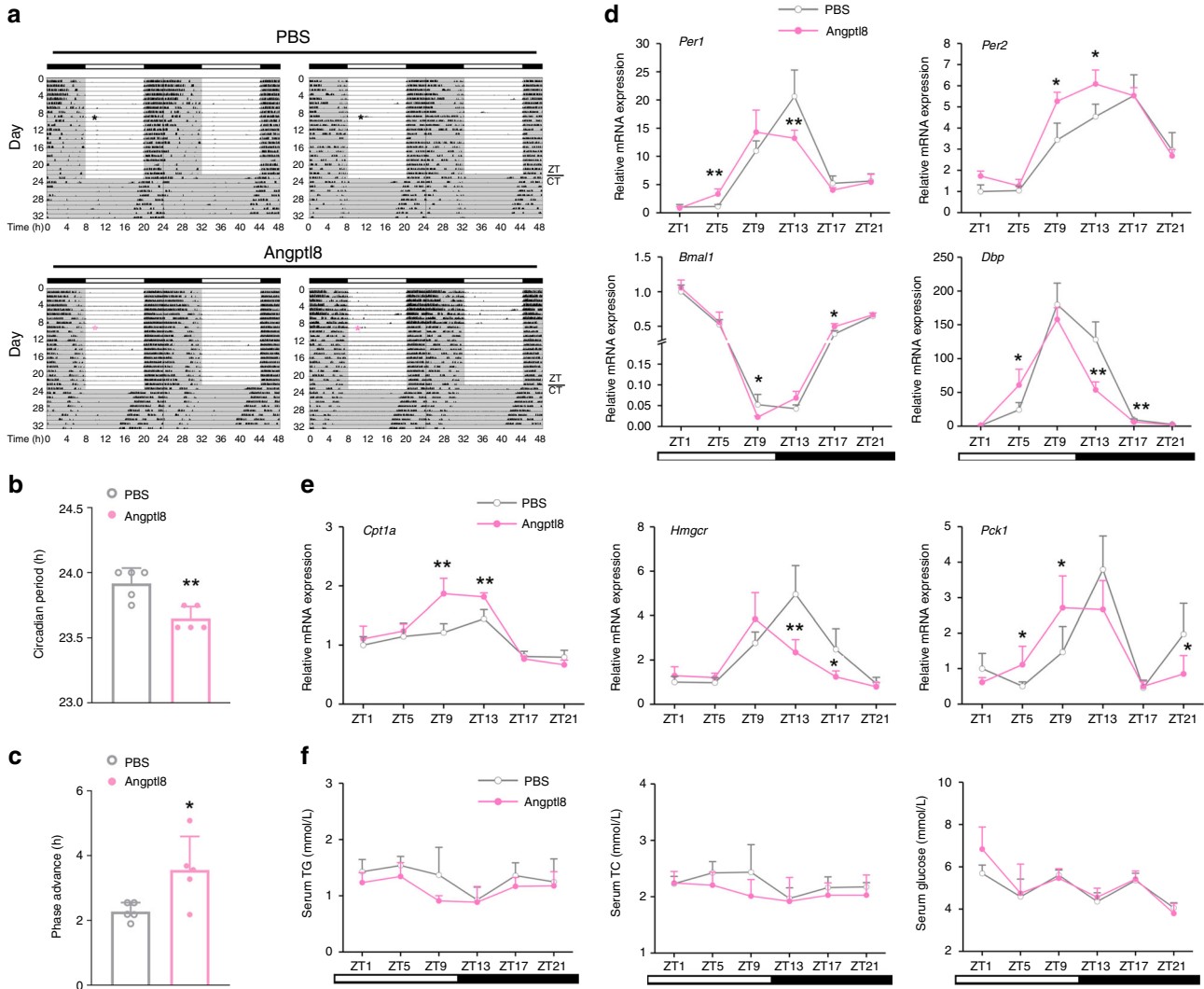

**Fig. 2** Angptl8 resets liver clock in vivo. **a** Representative actograms of the wheel-running activities of mice i.p. injected recombinant Angptl8 or vehicle (PBS) for 24 consecutive days at ZT2 (or CT2 when mice switched to DD). $n = 5$. **b** Calculation of mouse circadian period in **a**. **c** Calculation of mouse circadian phase shift in **a**. *$P < 0.05$ and **$P < 0.01$ Angptl8 vs. PBS, one-way ANOVA followed by Bonferroni's post hoc test. **d**, **e** qPCR analyses of clock and metabolic genes expression in the liver of mice receiving 15-day injection with recombinant Angptl8 or vehicle (PBS) at ZT2. $n = 5$, *$P < 0.05$ and **$P < 0.01$ Angptl8 vs. PBS at the same time point, two-way ANOVA followed by Bonferroni's post hoc test. **f** Serum levels of TG, TC and glucose in **d**. All data are presented as the means ± SD. Source data are provided as a Source Data file

**Angptl8 induces Per1 expression in vitro and in vivo.** Since Angptl8 may regulate the liver clock, we next elucidated the underlying mechanism. The acute induction of Per1/2 plays an important role in resetting the circadian clock of various tissues[34,35]. However, Angptl8 shock in mouse Hepa1c1c-7 hepatoma cells for 2 h induced the expression of key clock regulators, whereas *Per1* induction was the most significant compared to other factors (Fig. 4a). In addition, *Per1* induction at mRNA and protein levels was both dose and time-dependent in response to Angptl8 shock (Fig. 4b, c, Supplementary Fig. 4a and 4b). Consistently, the transcriptional activity of the proximal region of the *Per1* promoter, as well as *Per1* mRNA accumulation, was increased by 2 h of Angptl8 stimulation (Fig. 4d, e). Immunocytochemistry (ICC) analysis indicated that the cytoplasmic fraction of Per1 protein was also increased (Fig. 4f).

To confirm the inducible effect of Angptl8 on Per1 expression in vivo, we performed an acute injection of Angptl8 purified protein at ZT2 (the nadir of endogenous Angptl8), respectively. RT-qPCR and Western blot analyses revealed that hepatic Per1 expression was robustly increased by Angptl8 injection (Fig. 4g,

Supplementary Fig. 4c). Conversely, ZT7 refers to the time point that mice begin to enter the active phase and intake food after physiological fasting. In contrast, *Per1* mRNA expression levels were decreased to half at 30 min after an acute i.p. injection of the Angptl8-neutralized antibody (Anti-Angptl8) at ZT7 (the time point when the mice enter the active phase, and the serum levels of Angptl8 begin to rise), and a reduction in its protein levels occurred at 1 h later (Fig. 4h, Supplementary Fig. 4d).

**Kinases activation mediates Angptl8-induced Per1 expression.** To elucidate the molecular mechanisms by which the Angptl8 regulates Per1 gene expression in nucleus, we examined the effects of Angptl8 on the phosphorylation of key factors, such as MAPK, NF-κB, AKT, and GSK-3β proteins. As shown in Fig. 5a, Angptl8 increased the phosphorylation levels of ERK1/2, P38, NF-κB, and AKT, while leaving the GSK-3β phosphorylation unaltered. Quantitative data for all proteins was presented in Supplementary Fig. 5. More importantly, SB203580 (P38 MAPK inhibitor), Bay11-7082 (NF-κB inhibitor) and Wortmannin (AKT inhibitor) abrogated the Angptl8-induced Per1 transcription and

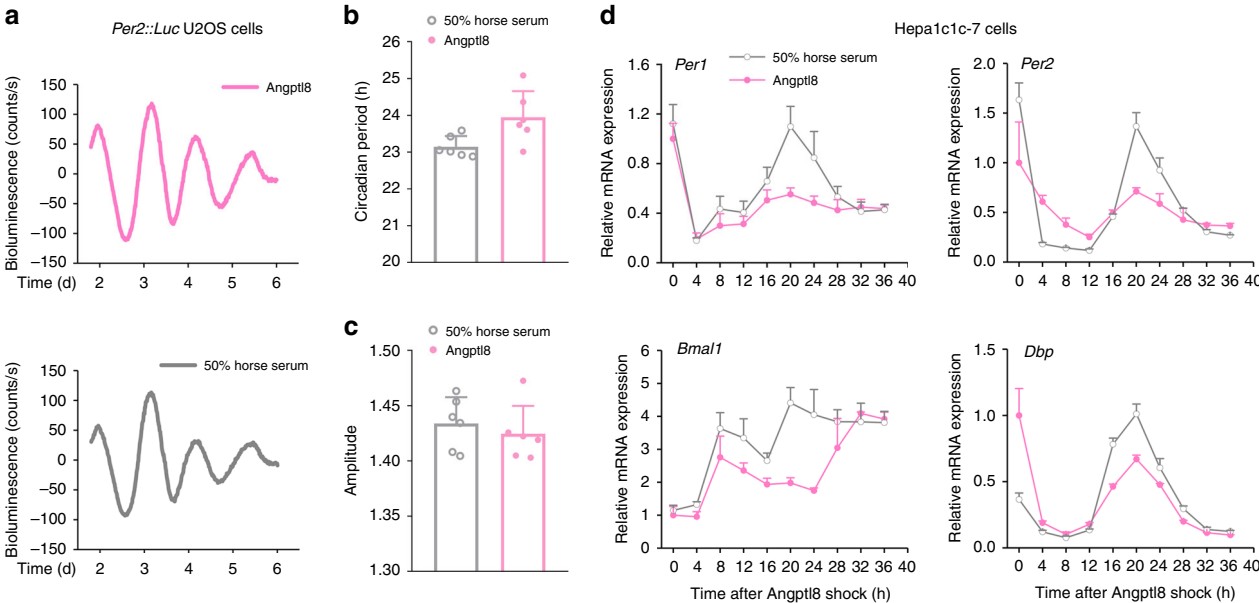

**Fig. 3** Angptl8 resets liver clock in vitro. **a** Representative luminescence traces of *Per2::Luc* U2OS cells stimulated with either Angptl8 or 50% horse serum for 2 h. **b**, **c** The periods and amplitudes of circadian transcriptional activities of *Per2* promoter in *Per2::Luc* U2OS cells stimulated with either 40 nM Angptl8 or 50% horse serum for 2 h. $n = 3$. All the data were calculated from fit curve data. **d** qPCR analyses of time-course expression of clock genes in mouse Hepa1c1c-7 cells similarly treated in panel **a**. $n = 3$. All data are presented as the means ± SD. Source data are provided as a Source Data file

translation, whereas U0126 (ERK1/2 inhibitor) failed to do that (Fig. 5b–e). These data suggest that P38 MAPK, NF-κB, and AKT mediate the actions of Angptl8 on Per1 expression.

**Per1 deficiency impairs the resetting functions of Angptl.** To detect the role of Per1 in mediating the function of Angptl8, we constructed a Per1 stable knockout (KO) Hepa1c1c-7 cell line by using a Crispr-Cas9 approach (the validation of KO efficiency is presented in Supplementary Fig. 6a and 6b). Although 2 h of Angptl8 shock induced robust oscillations of key clock, *Per1* deficiency attenuated the rhythmic amplitude of *Per2*, while increased the amplitudes of *Bmal1* and *Dbp*. Significant phase delays of these clock genes were observed in *Per1* KO cells (Fig. 6a and Supplementary Table 7). In addition, the time-course expression of metabolic genes was dampened in *Per1* KO Hepa1c1c-7 cells (Fig. 6b). To detect the role of *Per1* in the Angptl8-induced resetting of clock gene oscillation in vivo, we transduced adenoviruses expressing either scramble shRNA or *Per1* shRNA in the liver of Angptl8-treated mice at eighth day. The efficiency of shRNA-mediated Per1 knockdown was provided in Supplementary Fig. 6c. We found that the amplitude of clock genes, such as *Per2* and *Dbp*, was reduced, while leaving the circadian phases unchanged (Fig. 6c and Supplementary Table 8). Similar trends were observed in the amplitude of metabolic genes, including *Cpt1a*, *Hmgcr*, and *Pck1* (Fig. 6d). Taken the in vitro and in vivo observations together, we conclude that *Per1* deficiency impairs the resetting function of Angptl8 in the liver clock.

**Angptl8 binds to its receptor PirB.** Given that Angptl8 is a secreting hormone and is unable to penetrate cell membrane and regulate clock machinery, we examined whether Angptl8 delivers resetting signals through the cell membrane. Previous studies have demonstrated that the human immune inhibitory receptor leukocyte immunoglobulin-like receptor B2 (LILRB2) and its mouse ortholog paired immunoglobulin-like receptor (PirB) are receptors for several angiopietin-like proteins[23,36]. Specifically, Angptl8 reverses established cardiomyopathy by the activation of the PirB receptor on the cell membrane of resident adult cardiac

progenitor cells[37]. Therefore, we hypothesized that PirB is a potential receptor for Angptl8 in the liver. First, we assessed the mRNA expression of PirB and found that this protein was not rhythmically expressed in the mouse liver (Fig. 7a). Additionally, Angptl8 shock did not influence the mRNA and protein expression levels of PirB in mouse hepatoma cells (Fig. 7b, c). However, co-immunoprecipitation (Co-IP) assays demonstrated that these two proteins showed physical binding (Fig. 7d, e). ICC analysis confirmed that endogenous PirB and Angptl8 co-localized at the cell membrane (Fig. 7f).

**PirB mediates Angptl8-induced Per1 activation.** To get further insights into the role of PirB in the induction of Per1 expression by Angptl8, we specifically knocked down PirB expression in Angptl8-treated Hepa1c1c-7 cells by transducing adenoviruses expressing PirB shRNA. The knockdown of PirB impaired Angptl8-induced Per1 upregulation both at transcriptional and translational levels (Fig. 8a–c) and dampened clock gene oscillation (Fig. 8d and Supplementary Table 9). The Angptl8-induced phosphorylation of P38 MAPK, NF-κB and AKT, as the upstream regulators of Per1 expression, was also correspondingly blocked (Fig. 8e and Supplementary Fig. 7a–7c). Next, we treated cell with Angptl8 for 0, 5, and 10 min, respectively, and performed Co-IP analysis by using anti-PirB antibody. Our results showed that treatment of Angptl8 induced a rapid and transient reduction in PirB tyrosine phosphorylation, accompanied with a decrease in the association with SHP-1/2 (Fig. 8f and Supplementary Fig. 7d–7g). These data clearly suggest that Angptl8 decreased the auto-phosphorylation of PirB at tyrosine sites and thus reducing the recruitments of SHP1/2, and finally activated Per1 expression.

**Angptl8 knockdown dampens the liver clock reset by food.** The present data showed that Angptl8 treatment resets liver circadian rhythms, and Angptl8 is regulated by nutritional signals, indicating that this protein may be involved in the food-driven resetting of the liver clock and metabolic machinery. Notably, short-term refeeding (10 min) potently increases the hepatic mRNA and protein expression levels, as well as the serum

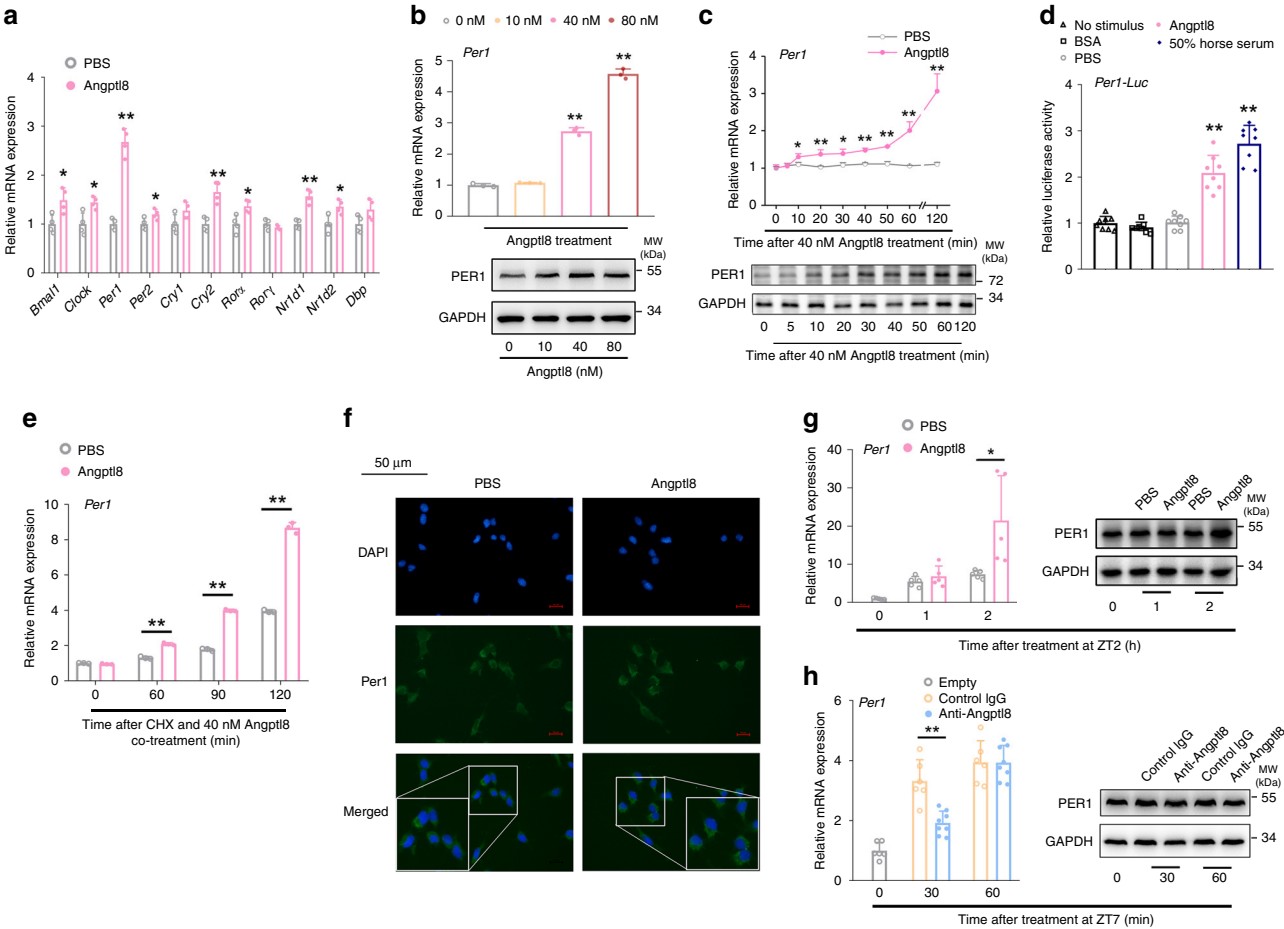

**Fig. 4** Angptl8 induces Per1 expression in vitro and in vivo. **a** qPCR analyses of the mRNA expression of core clock genes in mouse Hepa1c1c-7 hepatoma cells stimulated with either 40 nM Angptl8 or vehicle (PBS) for 2 h. $n = 3$, *$P < 0.05$ and **$P < 0.01$ Angptl8 vs. PBS, one-way ANOVA followed by Bonferroni's post hoc test. **b** qPCR and Western blot analyses of Per1 expression in Hepa1c1c-7 cells stimulated with Angptl8 at indicated doses for 2 h. $n = 3$, **$P < 0.01$ Angptl8 vs. PBS, one-way ANOVA followed by Bonferroni's post hoc test. **c** qPCR and Western blot analyses of Per1 expression in Hepa1c1c-7 cells stimulated with 40 nM Angptl8 for indicated times. $n = 3$, *$P < 0.05$ and **$P < 0.01$ vs.0 h, one-way ANOVA followed by Bonferroni's posthoc test. **d** Reporter gene assays in Hepa1c1c-7 cells transfected with plasmids expressing Per1 promoter-luc (−2100 to +100 bp) for 36 h and then treated with 40 nM Angptl8, 50% horse serum, or indicated controls for another 2 h. $n = 8$, **$P < 0.01$ vs. No stimulus group, one-way ANOVA followed by Bonferroni's post hoc test. **e** Per1 mRNA accumulation was assessed by CHX (cycloheximide) chase experiments. $n = 3$, **$P < 0.01$ Angptl8 vs. PBS, one-way ANOVA followed by Bonferroni's post hoc test. **f** ICC analysis of Per1 intracellular localization in Hepa1c1c-7 cells stimulated with Angptl8 or PBS for 2 h. **g** qPCR and Western blot analyses of Per1 expression in livers of mice subjected to an acute injection of recombinant Angptl8 or vehicle (PBS) at ZT2. $n = 5$, *$P < 0.05$ Angptl8 vs. PBS at the same time point, one-way ANOVA followed by Bonferroni's post hoc test. **h** qPCR and Western blot analyses of Per1 expression in the liver of mice subjected to an acute injection of Angptl8-neutralized antibody (50 μg/25 g, $n = 8$) or control IgG ($n = 6$) at ZT7. **$P < 0.01$ Anti-Angptl8 vs. control IgG at the same time point, one-way ANOVA followed by Bonferroni's post hoc test. All data are presented as the means ± SD. Source data are provided as a Source Data file

concentration, of Angptl8 (Supplementary Fig. 8a–c), suggesting that the hepatokine Angptl8 is rapidly responsive to food.

To further examine the liver-specific role of Angptl8 in the regulation of hepatic clock, we constructed mice with liver-specific Angptl8 knockdown by using AAV8 system carrying an Angptl8 shRNA. As shown in Supplementary Fig. 9a and Fig. 9a, AAV8-mediated knockdown of Angptl8 successfully blocked the refeeding-induced Angptl8 expression and secretion. To test whether postprandial Angptl8 induction is sufficient to influence liver clock gene expression, we analyzed the Per1/2 expression in the liver of these mice after fasting-refeeding. We found that Angptl8 shRNA successfully blocked both Per1 and Per2 mRNA expression in the mouse liver, even when the food was available again (Fig. 9b, c). Of note, it has been reported that various hormones, such as glucagon, insulin, GLP-1, OXM, and ghrelin, play a certain role in the resetting of circadian clock[38–42]. Also, these hormones can be regulated by the fasting/refeeding cycles,

raising the possibility that they may be involved in the Angptl8-induced liver clock re-entrainment. In our findings, knockdown of Angptl8 did not alter mouse serum levels of these hormones in response to the refeeding signals, suggesting that these hormones are not involved in the Angptl8-mediated activation of hepatic Per expression in response to food (Supplementary Fig. 9b–f).

To further investigate the effects of Angptl8 on the resetting of clock and metabolic gene oscillation in mouse liver in response to fasting/refeeding cycles, we subjected these mice to DD for 7 days, and fasted the mice for 12 h followed by refeeding for 21 h (the experimental flowchart was shown in Fig. 9d). As shown in Fig. 9e, f, Angptl8 shRNA constitutively decreased the hepatic expression and serum levels of Angptl8 in mice after refeeding. More importantly, refeeding increased the amplitude of Per2 oscillation, whereas decreased the amplitude of Bmal1 and Dbp oscillation, compared to that in the fasted cohort. In addition, refeeding signals also robustly induced phase advance of these

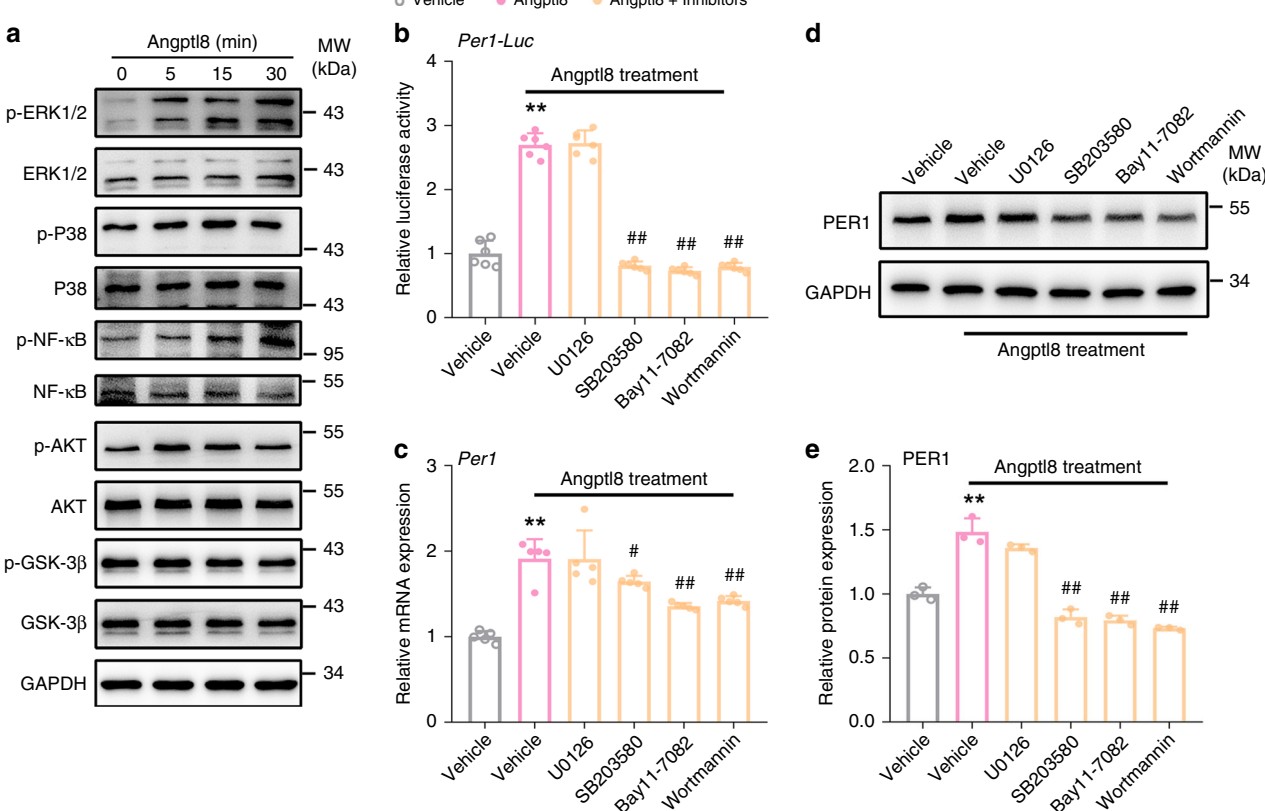

**Fig. 5** Kinases activation mediates Angptl8-induced Per1 expression. **a** Western blot analyses of ERK1/2, P38, NF-κB, AKT, and GSK-3β phosphorylation in Hepa1c1c-7 cells stimulated with 40 nM Angptl8 for indicated times. **b** Reporter gene assays in Hepa1c1c-7 cells transfected with plasmids expressing *Per1-Luc* for 36 h and then treated with 40 nM Angptl8 alone or in combinations of various signaling pathway inhibitors for 2 h. $n = 6$, **$P < 0.01$ Angptl8 vs. Vehicle group, ##$P < 0.01$ Angptl8 + inhibitor *vs.* Angptl8 alone group, one-way ANOVA followed by Bonferroni's post hoc test. **c**, **d** qPCR and Western blot analyses of Per1 expression in Hepa1c1c-7 cells treated with 40 nM Angptl8 alone or in combinations of various signaling pathway inhibitors for 2 h. $n = 3$, **$P < 0.01$ Angptl8 vs. Vehicle group, #$P < 0.05$ and ##$P < 0.01$ Angptl8 + inhibitor *vs.* Angptl8 alone group, one-way ANOVA followed by Bonferroni's post hoc test. **e** Quantitative analysis of Per1 protein levels in **d**. $n = 3$, **$P < 0.01$ Angptl8 vs. Vehicle group, ##$P < 0.01$ Angptl8 + inhibitor vs. Angptl8 alone group, one-way ANOVA followed by Bonferroni's *posthoc* test. All data are presented as the means ± SD. Concentrations: U0126, 10 μM; SB203580, 10 μM; Bay11-7082, 40 μM; Wortmannin 10 μM. Source data are provided as a Source Data file

clock gene rhythmicity. Contrasting, liver-specific Angptl8 knockdown attenuated the amplitude of *Per2*, while restored the amplitude of *Bmal1* and *Dbp* fluctuation. At the meanwhile, both manipulations regulated the phase of *Bmal1* rhythmicity in the liver of refed mice towards to that of the fasted cohort (Fig. 9g and Supplementary Table 10). Regarding the metabolic genes, the refeeding-induced pulse of *Hmgcr* oscillation was severely impaired when Angptl8 was knocked down (Supplementary Fig. 9g). Coincided with the findings in Angptl8 KO mice, liver-specific Angptl8 knockdown led to the decrease in serum TG levels of refed mice (Fig. 9h).

**Angptl8 neutralization dampens the liver clock reset by food.** To examine whether circulating Angptl8 induction mediates food-induced *Per1/2* upregulation in the liver, we quenched Angptl8 expression by injecting Anti-Angptl8 into refed mice after 12-h fasting. The neutralization of Angptl8 successfully blocked both *Per1* and *Per2* mRNA expression in the mouse liver, even when the food was available again (Fig. 10a, b). In contrast, Anti-Angptl8 treatment did not alter the serum levels of glucagon, insulin, GLP-1, OXM, and ghrelin (Supplementary Fig. 10a–10e). To further investigate the effects of circulating Angptl8 on the resetting of clock and metabolic gene oscillation in mouse liver in response to fasting/refeeding cycles, we housed mice in DD for 7 days and then subjected them to a 12-h fasting followed

by refeeding for 21 h. One hour before refeeding (at CT0), an acute injection of Angptl8-neutralized antibody (50 μg/25 g) or control IgG was administered to mice (the experimental flowchart was shown in Fig. 10c). We found that the neutralization of Angptl8 antagonized the effect of refeeding on the resetting of liver clock, evidenced by the decrease in the amplitudes of *Per2* and *Dbp* mRNA oscillation and the delay in the phase of *Bmal1* rhythmicity (Fig. 10d and Supplementary Table 11). Regarding the metabolic genes, the refeeding-induced pulse of *Hmgcr* oscillation was severely impaired when Angptl8 was blocked (Supplementary Fig. 10f). The diurnal oscillation of serum levels of metabolites, especially TG, was also dampened in refed mice (Fig. 10e). These findings collectively suggest that circulating Angptl8 is a critical mediator in the food-entrained resetting of the liver clock.

## Discussion

Food intake is an important Zeitgeber for the entrainment of circadian clocks in peripheral tissues, while the SCN clock is largely controlled by the LD cycle[43]. The sensitivity of peripheral clocks to food signals instead of light may help mammals more reasonably adapt to the environment, because the availability of food in nature does not always coincide with other external factors[44,45]. However, the uncoupling of peripheral clocks from the central clock when eating occurs during the resting phase

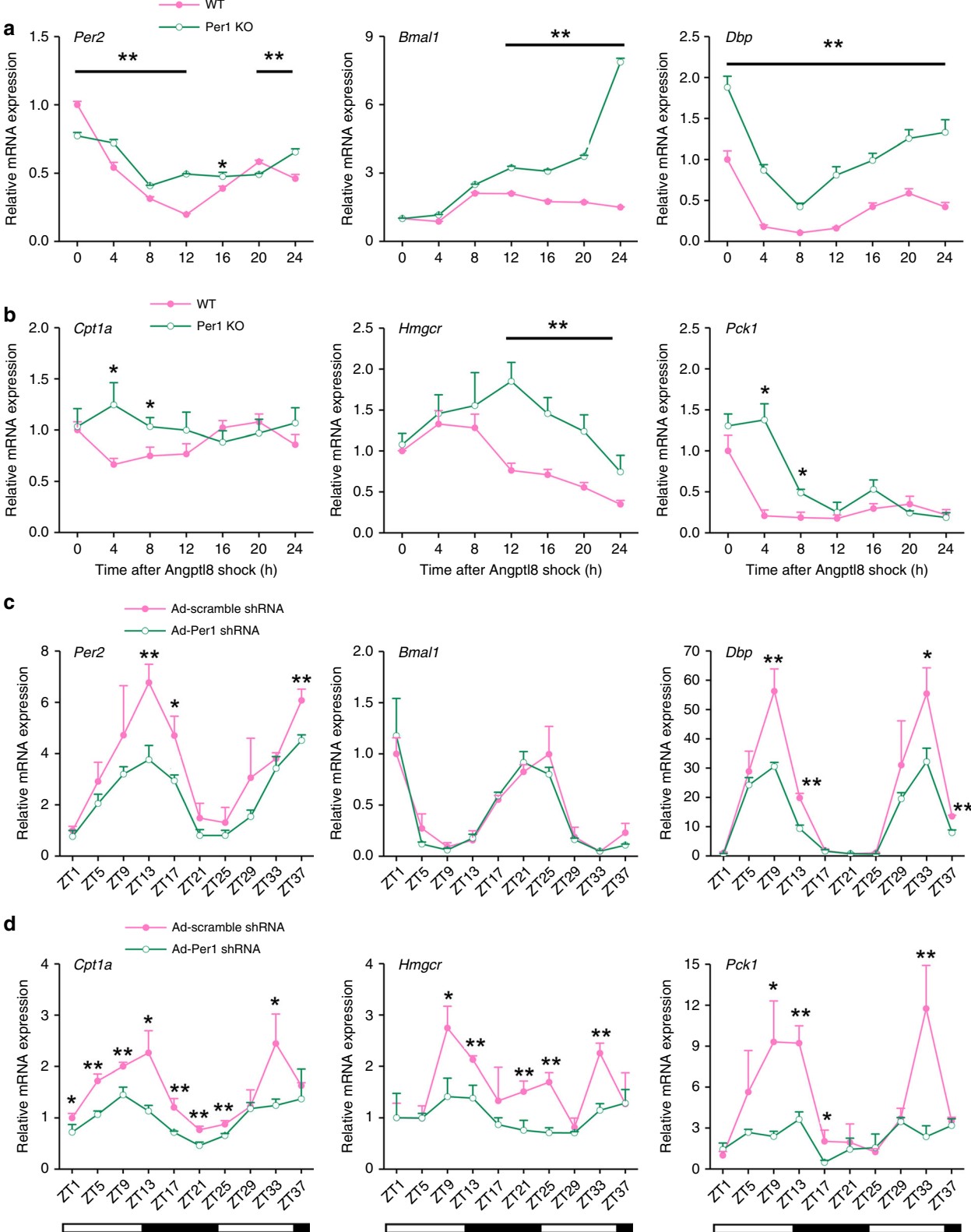

**Fig. 6** Per1 deficiency impairs the resetting functions of Angptl8. **a**, **b** qPCR analyses of time-course expression of clock and metabolic genes in *Per1* KO Hepa1c1c-7 hepatoma cell lines after Angptl8 shock. $n = 3$, *$P < 0.05$ and **$P < 0.01$ *Per1* KO vs. wild type (WT) cells at the same time point. On the other hand, mice were transduced with adenovirus expressing either scramble shRNA or Per1 shRNA through tail-vein injection at the eighth day during the Angptl8 injection. **c**, **d** qPCR analyses of clock and metabolic genes expression in the liver of mice receiving 15-day injection with recombinant Angptl8 or vehicle (PBS) at ZT2. $n = 3$, *$P < 0.05$ and **$P < 0.01$ Ad-Per1 shRNA *vs.* Ad-scramble shRNA at the same time point, two-way ANOVA followed by Bonferroni's post hoc test. All data are presented as the means ± SD. Source data are provided as a Source Data file

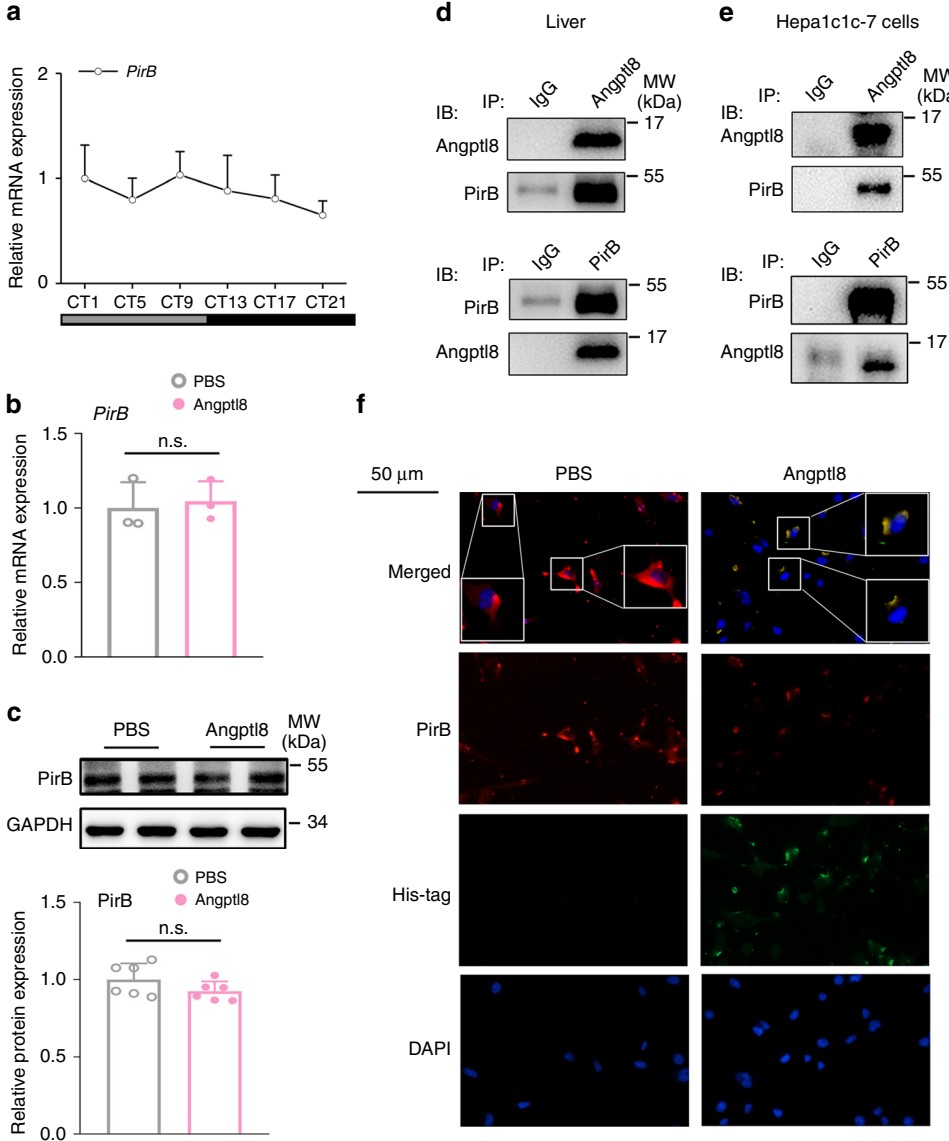

**Fig. 7** Angptl8 binds to its receptor PirB. **a** qPCR analyses of *PirB* mRNA expression in the liver of mice subjected to DD. $n = 8$. **b**, **c** qPCR and western blot analyses of PirB expression. Hepa1c1c-7 cells were stimulated with either 40 nM Angptl8 or PBS for 2 h. $n = 3$. **d** Co-IP assays in the mouse liver lysates. Immunoblots of precipitated proteins were performed by using indicated antibodies. **e** Co-IP assays in Hepa1c1c-7 cells cotransfected with plasmids encoding mouse Angptl8 and PirB. Immunoblots of precipitated proteins were performed by using indicated antibodies. **f** ICC analysis of colocalization of Angptl8 and PirB on the membrane of Hepa1c1c-7 cells treated with exogenous recombinant His-Angptl8 or vehicle (PBS) for 20 min. Source data are provided as a Source Data file

leads to internal circadian desynchrony, which is considered a risk factor to induce metabolic disorders[10]. Among peripheral clock systems, the liver clock is one of the most vital clock systems that can be reset by numerous food-related metabolites, such as glucose, insulin, and oxyntomodulin[39,41,46]. However, the food entrainment of the liver clock does not fully depend on these factors, and the global regulation of food-induced liver clock resetting has not been investigated. In the present study, we identified the secreting hepatokine Angptl8 as a potential direct link between food intake and hepatic clock resetting and metabolic gene transcription. Angptl8 was rhythmically expressed in the liver and induced by refeeding after overnight fasting. The injection of Angptl8 purified protein in mice at ZT2 advanced the oscillation phases of major clock and metabolic genes in the liver, while the blockade of Angptl8 in refed mice by either AAV-mediated shRNA delivery or a neutralized antibody caused the changes of clock and metabolic gene oscillation toward that

occuring in the fasted cohort. Mechanically, Angptl8 bound to PirB receptor at the cell membrane and acutely induced *Per1* and *Per2* expression at the transcriptional level. More importantly, the inhibition of Angptl8 signaling blocked the food-induced entrainment of the liver clock (Fig. 4f). Collectively, these findings add important information on molecular mechanism underlying food entrainment of the liver clock.

Angptl8 is a versatile secreted protein and plays important roles in diverse physiological processes. It is critical to select an appropriate animal model to elucidate tissue-specific functions of Angptl8. Angptl8 KO mice have been generated for more than 5 years[47], and potentially become a candidate to establish the tool platform in our present study. However, it does not escape from our notice that Angptl8 KO mice manifest various metabolic phenotypes, such as the significant decrease in the fat mass at the age of 12 weeks[47]. The changes in adipose tissues, a critical endocrine organ, will lead to the alteration of circulating

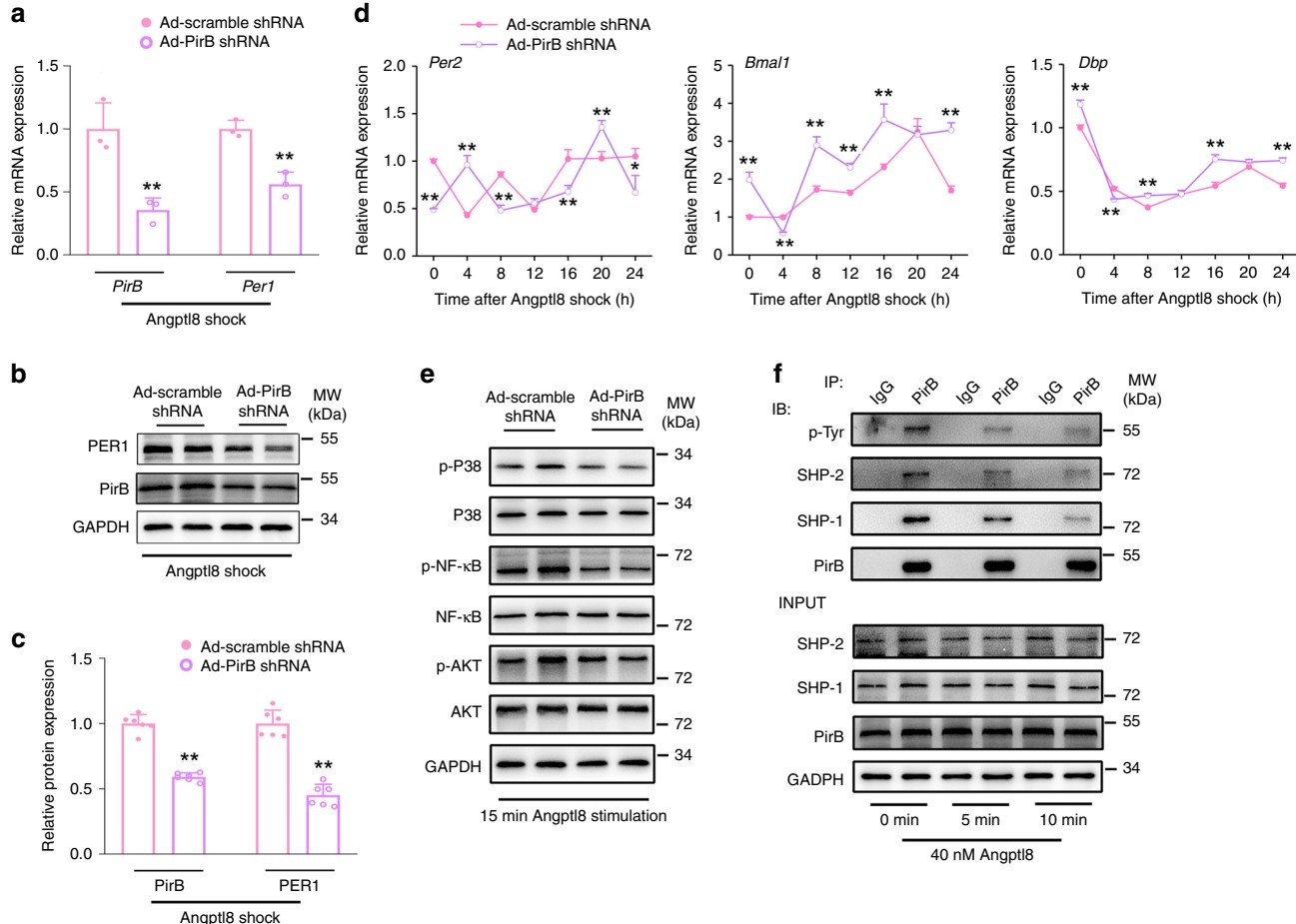

**Fig. 8** PirB mediates Angptl8-induced Per1 activation. **a, b** qPCR and Western blot analyses of PirB and Per1 expression in Hepa1c1c-7 cells transduced with adenoviruses expressing either PirB shRNA or scramble shRNA for 36 h followed by a 2-h Angptl8 shock. $n = 3$. **c** Quantitative analysis of PirB and Per1 protein levels. $n = 6$. For **a–c** **$P < 0.01$ Ad-PirB shRNA vs. Ad-scramble shRNA, one-way ANOVA followed by Bonferroni's post hoc test. **d** qPCR analyses of time-course expression of clock genes. $n = 3$, *$P < 0.05$ and **$P < 0.01$ Ad-PirB shRNA vs. Ad-scramble shRNA at the same time point, two-way ANOVA followed by Bonferroni's post hoc test. **e** Western blot analyses of P38, NF-κB, and AKT protein phosphorylation. Hepa1c1c-7 cells were transduced with adenoviruses expressing either PirB shRNA or scramble shRNA for 36 h followed by a 15-min Angptl8 stimulation. **f** Assessment of tyrosine phosphorylation of PirB protein and interactions between PIR-B and SHP-1/2 by using Co-IP assay. Hepa1c1c-7 cells were stimulated with either 40 nM Angptl8 for indicated times. Immunoblots of precipitated proteins were performed by using indicated antibodies. All data are presented as the means ± SD. Source data are provided as a Source Data file

adipokine levels, which has a potential impact on the homeostasis of liver clock (e.g. adiponectin induces phase advance of the liver clock in KK/Ta mice)[48]. In addition, the physiological roles of Angptl8 in the central nervous system (CNS) are still unknown. In our study, although injection of Angptl8 in LD cycles indeed shortened the circadian period, its effects on the period were relatively minor and may be secondary due to other metabolites which were produced in response to Angptl8 injection and were able to penetrate blood-brain barrier. Therefore, we could not exclude the possibility that the whole-body knockout of Angptl8 may influence the CNS and/or cause compensatory changes in mice, and indirectly regulates the liver clock. Therefore, although Angptl8 KO mice may serve as a feasible animal model in the present study, it is not a clean setting and possibly mask the direct regulation of liver clock by Angptl8. To overcome this problem and to authentically identify the liver-specific role of Angptl8 in resetting the peripheral clock, we constructed mice with liver-specific Angptl8 knockdown by using the AAV8 system harboring an Angptl8 shRNA, as well as quenched the circulating Angptl8 by using an Angptl8-neutrlized antibody in mice.

As acute responsive genes, *Per1/2* play an important role in the entrainment of circadian clock systems. The transient induction of *Per1/2* is involved in resetting the circadian clock in various tissues, including the SCN[49,50]. *Per1/2* expression was also upregulated in fibroblasts and hepatoma cells treated with DEX and serum shock[32,33]. In addition, *Per1/2* were critical in mediating the resetting of the SCN clock induced by the nonphotic Zeitgeber, melatonin[51]. Consistent with these findings, both *Per1/2* were induced after Angptl8 treatment, while *Per1* induction was more robustly induced in vivo, suggesting that additional mechanisms may finely and differently regulate *Per1* and *Per2*. Similarly, refeeding also sensitively induces hepatic *Per1/2* expression, which has been confirmed as an inalienable part of the food-evoked clock resetting mechanism. Furthermore, either AAV-mediated liver-specific Angptl8 knockdown or blockade of Angptl8 in circulation by using a neutralized antibody in refed mice after overnight fasting suppressed the upregulation of hepatic *Per1/2* expression. Thus, we conclude that food intake stimulates Angptl8 secretion and increases circulating levels of Angptl8, which resets the liver clock via the induction of the core

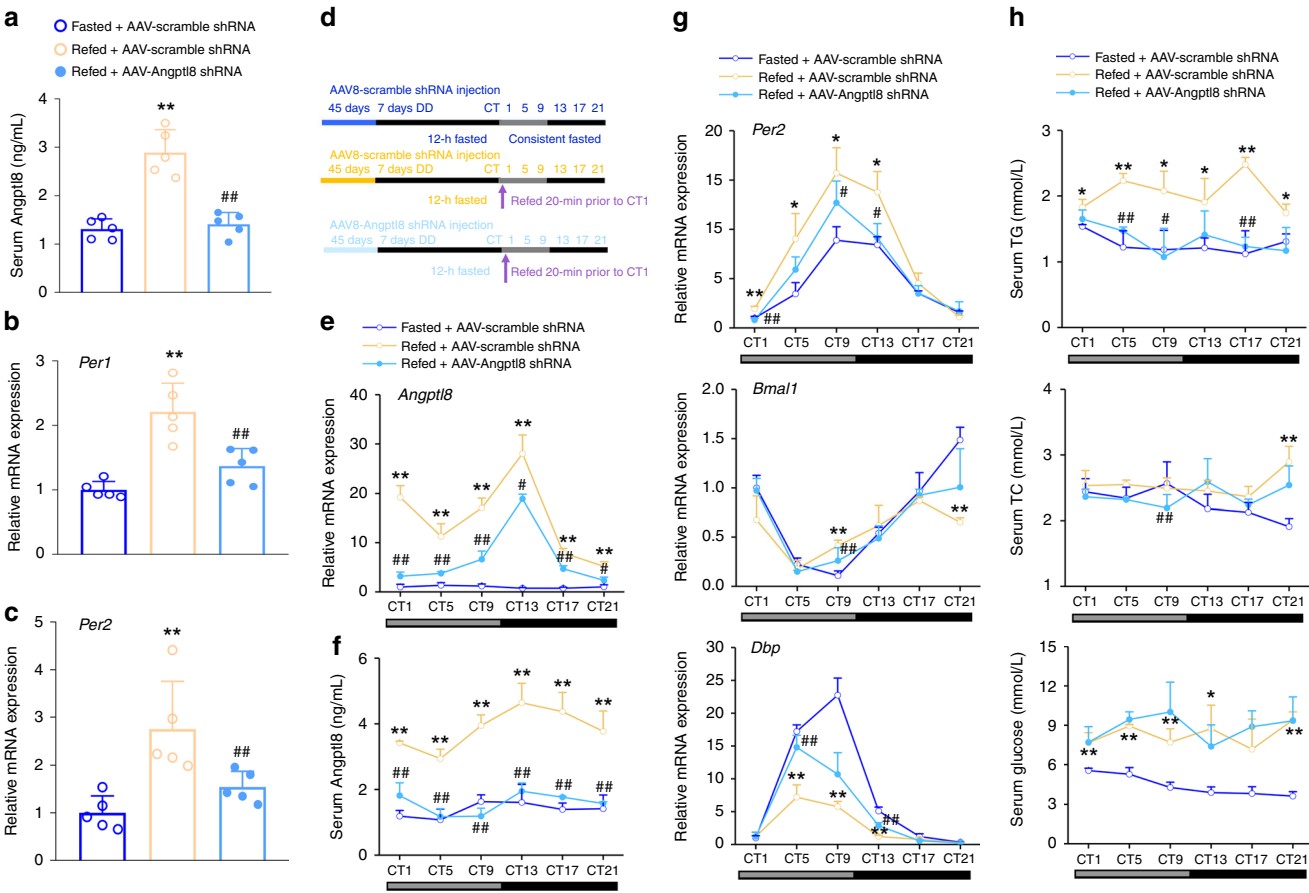

**Fig. 9** Angptl8 knockdown dampens the liver clock reset by food. Mice were transduced with AAVs carrying either Angptl8 shRNA or scramble shRNA for 45 days. **a** Serum Angptl8 levels in mice subjected to 12-h food deprivation (ZT13-1) or 12-h food deprivation followed by 20 min refeeding. $n = 5$, **$P < 0.01$ Refed vs. Fasted mice, ##$P < 0.01$ AAV-Angptl8 shRNA *vs.* AAV-scramble shRNA, one-way ANOVA followed by Bonferroni's post hoc test. **b**, **c** qPCR analyses of *Per1* and *Per2* mRNA expression in the liver of mice from **a**. $n = 5$, **$P < 0.01$ Refed vs. Fasted mice. ##$P < 0.01$ AAV-Angptl8 shRNA vs. AAV-scramble shRNA, one-way ANOVA followed by Bonferroni's post hoc test. **d** A schematic diagram illustrating the time design of DD and a fasting/refeeding cycle for liver-specific Angptl8 knockdown mice. **e**, **f** Time-course mRNA and serum levels of Angptl8 in mice subjected to 7-day DD and fasted for 12 h at CT13 or 12-h food deprivation followed by refeeding for 21 h. **g** qPCR analyses of clock genes. **h** Serum levels of certain metabolites. For **e**–**h** $n = 3$, *$P < 0.05$ and **$P < 0.01$ refed vs. fasted mice. #$P < 0.05$ and ##$P < 0.01$ AAV-Angptl8 shRNA vs. AAV-scramble shRNA, two-way ANOVA followed by Bonferroni's post hoc test. All data are presented as the means ± SD. Source data are provided as a Source Data file

clock genes *Per1/2*, and *Per1* functions more dominantly than *Per2*.

Given that Angptl8 is a secreted that is unable to penetrate cell membrane and regulate clock machinery, which is primarily localized in nuclei, the next important question is: how does Angptl8 deliver resetting signals through the cell membrane? Previous studies have demonstrated that the human LILRB2 and its mouse ortholog PirB are receptors for several angiopoietin-like proteins[36,52]. Specifically, Angptl8 reverses established cardiomyopathy by PirB receptor activation on the cell membrane of resident adult cardiac progenitor cells[37]. Therefore, we hypothesized that there is also a similar interaction between Angptl8 and PirB in hepatocytes. Indeed, although Angptl8 did not influence the expression of PirB in mouse Hepa1c1c-7 cells, Co-IP assays confirmed that these two factors physically bind in vivo and colocalize at the cell membrane. More importantly, the knockdown of PirB impaired Angptl8-induced *Per1* upregulation and dampened clock gene oscillation. These data clearly suggest that PirB, as a membrane receptor, at least partially, if not totally, mediates the resetting activity of Angptl8 in the liver clock. Notably, two membrane proteins, in addition to PirB, can also bind to Angptl8. One protein, Marco, is expressed in macrophages found in the pancreas and regulates lipid metabolism. The

other protein, RTN4R, is a neuronal receptor[53]. Whether these two proteins are also involved in the resetting function of Angptl8 remains unknown.

Since Angptl8 increases Per1 expression and the signal transduction is dependent on the membrane receptor PirB, it is necessary to elucidate the molecular mechanisms by which the Angptl8-PirB axis regulates Per1 gene expression in nucleus. Previous studies have indicated that Angptl8 increases the insulin sensitivity in HepG2 cells through triggering the phosphorylation of AKT and GSK-3β proteins[54]. On the other hand, PirB, acting as a receptor, responds to the external inflammatory signals or bacterial stimulation and regulates the phosphorylation of downstream MAPK and NF-κB factors through recruitment of SH2 containing phosphatases (e.g. SHP-1/2)[55–58]. Notably, these factors are intensively involved in the regulation of Per1 expression via various transcription factors including *c-Fos*, *FosB* and *c-Jun*[33,59]. These findings strongly suggest that various kinases (MAPK and AKT), as well as transcriptional factors (e.g. NF-κB), may mediate the regulation of Per1 expression by the Angptl8-PirB axis. Therefore, in the present study, we examined the changes of these kinases and transcriptional factors in response to Angptl8. Our results revealed that Angptl8-induced the phosphorylation of P38 MAPK, NF-κB, and AKT mediated its action

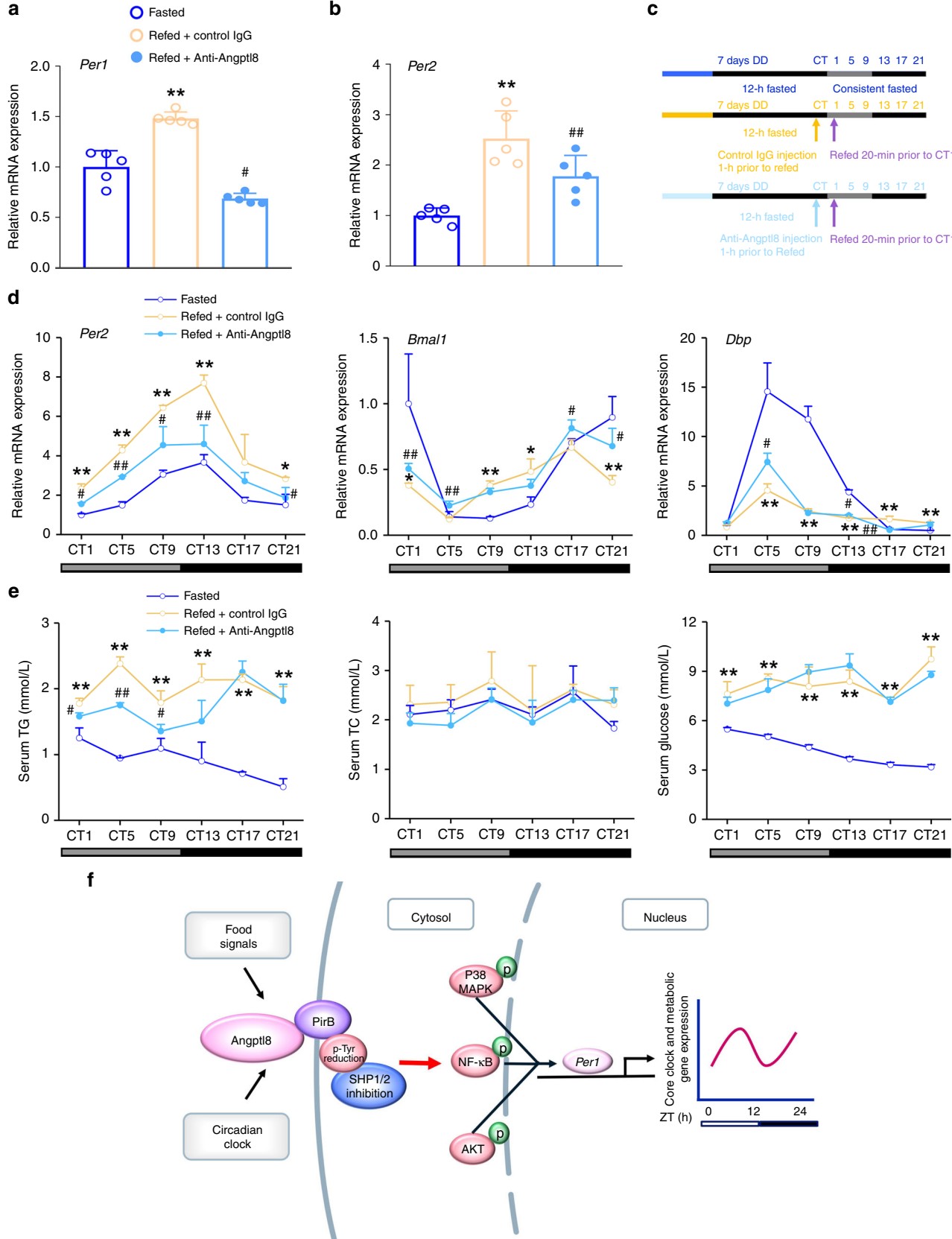

**Fig. 10** Angptl8 neutralization dampens the liver clock reset by food. **a**, **b** qPCR analyses of *Per1* and *Per2* mRNA expression in the liver of mice subjected to 12-h food deprivation (ZT13-1) or 12-h food deprivation followed by 20 min refeeding. One hour before refeeding (at ZT0), an acute injection of Angptl8-neutralized antibody (50 μg/25 g) or control IgG was administered to mice. $n = 5$, $**P < 0.01$ Refed vs. Fasted mice. $^{#}P < 0.05$ and $^{##}P < 0.01$ Anti-Angpptl8 vs. control IgG, one-way ANOVA followed by Bonferroni's post hoc test. **c** A schematic diagram illustrating the time design of DD, a fasting/refeeding cycle and injection of Angptl8-neutralized antibody. **d** qPCR analyses of time course expression of clock genes in the liver of mice subjected to an acute injection of Angptl8-neutralized antibody (50 μg/25 g) or control IgG. **e** Serum levels of certain metabolites. For **d**, **e** $n = 3$, $*P < 0.05$ and $**P < 0.01$ Refed vs. Fasted mice, $^{#}P < 0.05$ and $^{##}P < 0.01$ Anti-Angpptl8 vs. control IgG at the same time point, two-way ANOVA followed by Bonferroni's posthoc test. All data are presented as the means ± SD. **f** A model illustrating the mediating role of Angptl8 in the food-driven resetting of the liver circadian clock in mice. Source data are provided as a Source Data file

on Per1 expression. More importantly, the knockdown of PirB impaired Angptl8-induced such phosphorylation. Co-IP analysis indicated that the Angptl8 decreased the auto-phosphorylation of PirB at tyrosine sites and reducing the recruitments of SHP1/2. Taken together, our experiments demonstrated that Angptl8 induced the phosphorylation of P38 MAPK, NF-κB, and AKT through decreasing the auto-phosphorylation of PirB at tyrosine sites and reducing the recruitments of SHP1/2, and finally activated Per1 expression.

Angptl8 is considered a novel but atypical member in the Angptl family because of its unique structure and crucial effects on lipid metabolism[24,60]. Additionally, Angptl8 participates in the pathogenesis of metabolic disorders, such as non-alcoholic fatty liver disease and diabetes[61,62]. The present study also revealed an unexpected role for Angptl8 in the resetting of the liver clock. Taken together, Angptl8 may serve as a critical integrator in coordinating the circadian clock and energy metabolism in the liver. At the metabolic level, recent studies have implied an essential role of Angptl8 in the regulation of TG homeostasis and VLDL secretion during the fasting-refeeding transition. Mechanistically, refeeding signals trigger the liver X receptor α (LXRα) and RXR to form a heterodimer, and further activating the transcription of *Angptl8* through binding to the LXR response elements on its proximal promoter. In contrast, fasting signals induce the accumulation of glucocorticoids in the circulating system. This accumulation activates and recruits glucocorticoid receptor (GR) dimmers to the nGRE located on the *Angptl8* promoter region, thus inhibiting *Angptl8* transcription[28]. Coincided with these findings, our present study indicated that the refeeding robustly increased *Angptl8* mRNA expression in 10 min (Supplementary Fig. 8), suggesting a rapid turnover of *Angptl8* mRNA in response to the fasting and refeeding transition. Indeed, the half-life of *Angptl8* mRNA is only 15.7 min and such an instability is important for its oscillation and function[28].

Compared with other known integrators of clock and metabolism (such as PGC-1α and PPAR family members), we propose that the administration of Angptl8 has the following advantages: (1) Angptl8 has no effect on the SCN master clock. In vivo injection of purified Angptl8 protein did not change the oscillation of core clock genes in the SCN, suggesting that Angptl8 functions exclusively in the periphery, leaving SCN untouched. (2) We carefully titrated the doses of Angptl8 in the in vivo injection and found that a low dose (1 μg/kg) of Angptl8 showed a modest effect on systemic metabolism but sufficiently regulated the liver clock. In contrast, previous studies on the roles of metabolic regulation of Angptl8 preferred using a adenovirus-mediated gain-of-function strategy, which led to an artificially high level of Angptl8[28]. Only in such cases can Angptl8 efficiently function in metabolic regulation. The difference in the window of manipulating Angptl8 levels when targeting clock and metabolism will enable flexibility in the modulation of these two pathways in a separate or combined manner. (3) As a secreted protein, Angptl8 is easily blocked by using a neutralized antibody. Thus, the modulation of Angptl8 is more convenient than that of nuclear factors.

Last but not least, our data indicated that Angptl8 is predominately secreted by the liver, and regulates the liver clock in turn, suggesting it is possible that this circulating hepatokine may transmit time information among different liver cells, such as hepatocytes, hepatic satellite cells and Kupffer cells. However, the physiological roles of Angptl8 in HSC and Kupffer cells currently remain unknown and is interesting to be investigated in the future study. On the other hand, it should be noted that Angptl8 is a secreted protein and can be delivered to various peripheral tissues through circulation. This factor also responds robustly to the external stimuli, in particular nutritional signals. Therefore, we believe that evaluation of the physiological functions of Angptl8 at the organ/tissue level (when specifically considering that the resetting of peripheral clock is largely dependent on the hormone cues balanced by the organ crosstalk) gets higher priority than that at the intra-tissue level.

In conclusion, we provided the evidence that Angptl8 is induced in circulation and in the liver in response to food intake. Functionally, this protein mediates the resetting of the liver clock by food through the PirB/Per1 axis. These interesting findings suggest that Angptl8 is a hepatokine in the orchestration of the liver clock, in addition to its diverse effects on metabolic regulation. Although other factors, such as insulin and glucocorticoids, also have similar actions, targeting these proteins may cause severe side-effects and clinical risks[63–65]. The present study provides a safer and more convenient drug target to treat metabolic disorders arising from circadian clock disruption from the view of chronotherapy.

## Methods

**Animals.** All animal procedures in this investigation conform to the Guide for the Care and Use of Laboratory Animals published by the US National Institutes of Health (NIH publication No. 85–23, revised 1996) and was approved by the Laboratory Animal Care & Use Committee at China Pharmaceutical University (Permit number SYXK-2016–0011). Male C57BL/6 J mice were maintained in a 12 h LD cycle and in a temperature- and humidity-controlled environment. Zeitgeber time zero (ZT0) referred to lights on. For fasting/re-feeding experiments, the mice were either fed ad libitum, or subjected to 16-h fasting (from ZT9 to ZT1 the next day) or 16-h fasting followed by 20-h re-feeding. For analysis of the circadian expression of Angptl8 in the liver, mice were maintained under LD for 1 week and subsequently subjected to DD for 6 weeks. For time-restricted feeding experiments, the mice were fed exclusively at night (night feeding from ZT12 to ZT0, NF) for 10 days, and then these animals were divided into two groups: one was maintained on NF, while the other group was switched to day feeding (DF, from ZT0 to ZT12). The mice were subjected to these feeding rhythms for 7 days and then sacrificed at ZT0 and ZT12. To investigate the effect of Angptl8 on resetting liver clock, the mice were *i.p.* injected with recombinant mouse His-Angptl8 protein (Cloud-Clone Corp., Wuhan, China) or its vehicle (PBS) at ZT2. In contrast, to knock down Angptl8 expression in liver, we transduced AAV8 system carrying shRNA against scramble (as a negative control) or Angptl8 (designed and synthesized by Hanbio, Shanghai, China) into mice at a dose of $1 \times 10^{12}$ vg through tail-vein injection. In addition, the immuno-neutralized Angptl8 antibody (synthesized by Bioworld Corp., Nanjing, China, 50 μg/25 g) or control IgG was i.p. injected into mice to quench Angptl8[66]. Besides, to knock down Per1 in mouse liver, we transduced adenoviruses expressing either scramble shRNA or Per1 shRNA (designed and synthesized by Genepharma, Shanghai, China) into mice at a dose of $1.5 \times 10^9$ plaque-forming units (PFU) per mouse through tail-vein injection. Detailed sequences for shRNA oligonucleotide sequences were listed in Supplementary Table 12. Animals were sacrificed at indicated time points by cervical dislocation.

**High-throughput RNA sequencing**. To screen out candidate genes that mediate the food-driven resetting of the liver clock, the mice were subjected to fasting/re-feeding cycles or DD for 6 weeks as described above. Liver samples were pooled, and total RNA was isolated for the construction of RNA-seq libraries. The quality of the RNA libraries was evaluated using the Agilent 2200 TapeStation (Agilent Technologies, USA). Library sequencing was performed on a HiSeq 3000 sequencing platform (Illumina Company, USA) by Guangzhou RiboBio Corp., China. Fragments Per Kilobase of transcript per Million mapped reads (FPKM) were calculated for additional statistics. All the reads were mapped to the mouse genome (GRCm38/mm10).

**Wheel-running activity**. Six-week-old mice were placed in standard mouse cages equipped with infrared sensors to detect locomotor activity. After 10-day accommodation to an LD cycle, the mice were i.p. injected daily with 1 μg/kg of recombinant Angptl8 or vehicle (PBS) for 24 constitutive days (at ZT2 for 15 days in LD and at CT2 for another 9 days when the mice were switched to DD). Wheel revolutions were recorded in 6-min bins by using ClockLab system (Actimetrics, Wilmette, IL, USA). The circadian period and phase shift in DD were determined using ClockLab analysis software (Actimetrics, Wilmette, IL, USA)[50,67].

**Cell culture and real-time monitoring assay**. Mouse Hepa1c1c-7 hepatoma cells (ATCC, Manassas, VA, USA) were cultured at 37 °C and 5% $CO_2$-95% air in α-MEM (Invitrogen, CA, USA) supplemented with 10% fetal bovine serum (FBS). To detect the circadian mRNA expression of clock genes in response to Angptl8, we treated the cells with Angptl8 for 2 h, and collected cell samples at 4-h intervals. For the real-time monitoring assay, human Per2::Luc U2OS cells (a gift from Zhang Eric Erquan[68], National Institute of Biological Sciences, Beijing, China) plated onto 35-mm dishes ($6 \times 10^5$ cells/dish) were cultured at 37 °C and 5% $CO_2$ −95% air in high-glucose DMEM supplemented with 10% FBS for 5 days, and then the culture medium was changed into serum-free DMEM for another 24 h. To study the synchronization effects of Angptl8, the cells were treated with Angptl8 or 50% horse serum (HyClone, Logan, UT, USA) for 2 h. The medium was subsequently replaced with recording medium (phenol red-free DMEM supplemented with 10% FBS, 10 mM HEPES buffer (pH 7.0), 0.1 mM luciferin (Promega, Madison, WI, USA), 100 units/mL penicillin and 100 μg/mL streptomycin)[69]. Baseline fluctuations of the raw data were fitted to polynomial curves and subtracted from the raw data by using the LM Fit (damped sine) tool of LumiCycle analysis software (Actimetrics, Wilmette, IL, USA). The period length and amplitude were calculated from the raw data derived from 48 to 120 h after synchronization by using MATLAB software (9.0 Version R2016a, MathWorks Inc., MA, USA).

**RT-qPCR analysis**. Total RNA was isolated using Trizol reagent (Invitrogen, USA), reverse transcribed, and analyzed by quantitative PCR (qPCR) using SYBR Green (Takara, Japan) and the LightCycler® 480 System (Roche, Basal, Switzerland). The primers for mouse 36B4 were included for normalization. A complete list of PCR primers is shown in Supplementary Table 12.

**Western blot analysis**. Liver tissues were homogenized, and the cells were lysed in RIPA buffer. The protein concentration was quantified with a BCA protein quantification kit (Bio-Rad, Hercules, CA, USA). Equal amounts of protein were loaded and separated by 10% SDS-PAGE and then transferred onto polyvinylidene difluoride membranes (Millipore, Bedford, MA, USA). The membranes were incubated overnight with appropriate primary antibodies. Bound antibodies were then visualized using HRP-conjugated secondary antibodies. A quantitative analysis was performed by using NIH ImageJ 1.32j software. The Western blot results are shown as representative blots from two animals randomly selected from each group. The anti-Per1 (Cat. No. 13463-1-AP, 1:1000 dilution) antibody was obtained from Proteintech (Chicago, IL, USA), the anti-Angptl8 (Cat. No. PA5-38043, 1:1000 dilution) and anti-PirB (Cat. No. MA5-24049, 1:500 dilution) antibodies were purchased from Thermo Fisher Scientific (USA), the anti-phospho-ERK1/2 MAPK (Thr 202/Tyr 204, Cat. No. 4377, 1:1000 dilution) and anti-total ERK1/2 MAPK (Cat. No. 9102, 1:1000 dilution), anti-phospho-P38 MAPK (Cat. No. 9212, 1:1000 dilution), anti-phospho-AKT (Ser 473, Cat. No. 4060, 1:1000 dilution), anti-total AKT (Cat. No. 9272, 1:1000 dilution), anti-phospho-GSK3β (Ser 9, Cat. No. 9323, 1:1000 dilution) and anti-total GSK3β (Cat. No. 9315, 1:1000 dilution), anti-SHP1 (Cat. No. 3759, 1:1000 dilution), anti-SHP2 (Cat. No. 3397, 1:1000 dilution) and anti-phospho-tyrosine (Cat. No. 9411, 1:2000 dilution) antibodies were obtained from Cell signaling technology (Shanghai, China), and anti-total-P38 MAPK (Thr 180/Tyr 182, Cat. No. 11253, 1:1000 dilution) were purchased from SABiosciences (MD, USA), anti-phospho-NF-κB (Ser 529, Cat. No. BS4317, 1:1000 dilution) and anti-total NF-κB (Cat. No. BS9879, 1:1000 dilution) were obtained from Bioworld (Nanjing, China), the anti-GAPDH (Cat. No. KC-5G5, 1:10000 dilution, Shanghai, China) antibody was obtained from Kangchen Biotech. Uncropped images are shown in Supplementary Fig. 11.

**Plasmids, transfection and reporter gene assays**. The Per1 promoter (−2100 to +100 bp) was amplified from mouse genomic DNA by using the primers shown in Supplementary Table 12. The sequences were validated by sequencing and cloned

into a PGL3-basic vector using the KpnI and MluI restriction sites. Plasmids encoding mouse full-length Angptl8 and the PirB CDS domain were synthesized by Bioworld Technology, Inc. (Nanjing, China). All transient transfections were conducted using Lipofectamine 3000 (Invitrogen, CA, USA) according to the manufacturer's instructions. For the luciferase reporter assays, 100 ng of reporter plasmids were transfected into mouse Hepa1c1c-7 cells. After 36 h, the cells were treated with 40 nM Angptl8, 50% horse serum (as a positive control) or two negative controls (PBS or BSA) for another 2 h. Relative luciferase activities were determined using a dual luciferase system (Promega, Madison, WI, USA).

**Immunocytochemistry**. To analyze the Per1 intracellular localization, mouse Hepa1c1c-7 cells were treated with or without Angptl8 for 2 h. Cells were then fixed in ice-cold 4% paraformaldehyde for 30 min, followed by blocking in 5% goat serum for 1 h, and then incubation with rabbit polyclonal anti-Per1 antibody overnight at 4 °C. To detect the colocalization of Angptl8 and PirB on the cell membrane, mouse Hepa1c1c-7 cells were treated with or without His-Angptl8 for 20 min. After washing, the cells were incubated with mouse monoclonal anti-His and rat monoclonal anti-PirB antibodies overnight at 4 °C. The cells were then probed with secondary antibodies conjugated to Alexa Fluor 488 (Cat. No. 4408 s, 1:500 dilution, Cell Signaling Technology) and Alexa Fluor 555 (Cat. No. 4413 s, 1:500 dilution, Cell Signaling Technology) for 1 h at room temperature[70]. Nuclei were identified with DAPI. Nonimmune IgG was used as a negative control. The sections were photographed with a Nikon fluorescence microscope (ECLIPSE, Ts2R-FL).

**Co-IP analysis**. Liver tissues were homogenized, and the cells were lysed in RIPA buffer. After centrifugation, 50 μg of protein lysate was incubated with 20 μL of Protein-A/G agarose beads (Roche, USA) and 2 μg of anti-Angptl8 or anti-PirB antibody. After overnight incubation, the immune complexes were centrifuged and washed four times with ice-cold IP wash buffer. The immunoprecipitated protein was further analyzed using Western blotting.

**Adenoviruses and Per1 knockout Hepa1c1c-7 cell line construction**. For PirB knockdown, adenoviruses expressing either scramble or shRNA targeting mouse PirB were designed, validated, and synthesized by Hanheng Biotech Corp. (Shanghai, China). Detailed shRNA oligonucleotide sequences were listed in Supplementary Table 12. Per1-deficient Hepa1c1c-7 cells were established using a CRISPR/Cas9 system according to the standard protocol provided by the Zhang lab[71]. Briefly, a single guide RNA (sgRNA) was generated by the online CRISPR Design Tool (http://crispr.mit.edu/) and cloned into SpCas9-2A-puro vector (PX459). After confirming the cutting efficiency of the sgRNA, the SpCas9 vectors were transfected into Hepa1c1c-7 cells. Sorted puromycin-resistant Hepa1c1c-7 cells were seeded onto 96-well plates for mono-colonization. After four weeks, the cell clones derived from single cells were detected for gene expression by DNA sequencing. The following sgRNA sequences targeting Per1 were used: sgRNA1: CTAGAAGGGGCCGATGGGGG and sgRNA2: AGGCCCGGAGAACCTTTTTG.

**Serological analysis**. Blood samples were collected in nonheparinized tubes and centrifuged at $1000 \times g$ for 10 min at 4 °C. The serum levels of Angptl8, TG, total TC, glucose, glucagon, insulin, glucagon-like peptide-1 (GLP-1), oxyntomodulin (OXM) and ghrelin were determined by using commercial kits (EIAab Science Corp., Wuhan, China for Angptl8, Millipore, Billerica, MA, USA for insulin, Sigma, St Louis, MO, USA for GLP-1 and ghrelin, Cusabio Corp., Wuhan, China for OXM, and the Jiancheng Institute of Biotechnology, Nanjing, Jiangsu, China for the other factors), according to the manufacturer's instructions. The blood glucose levels were measured by using a glucose monitor (Roche Diagnostics, Basel, Switzerland).

**Statistical analysis**. Statistical analysis was performed by using Origin 8 software (version 8.6, OriginLab Corporation, MA, USA). The data are presented as the means ± SD (standard deviation). Circadian variations, including amplitude and phase shift, were calculated by fitting a cosine-wave equation $[y = \text{baseline} + (\text{Amplitude} \times \text{Cos} (2 \times \pi \times (\text{x-phaseshift})/24)]$ on clock gene expression, with a fixed 24-h period (detailed data for the oscillation of clock genes were presented in Supplementary Table 5–11). Time series data were analyzed using one-way or two-way ANOVA followed by Bonferroni's post hoc test. A P-value of less than 0.05 was considered to be statistically significant. Unless otherwise indicated, the statistics was performed using Student's t-test when only two groups were compared.

**Reporting summary**. Further information on research design is available in the Nature Research Reporting Summary linked to this article.

## Data availability
The data supporting the findings are available within the article and Supplementary Information. RNA-seq data files have been deposited into Gene Expression Omnibus database (www.ncbi.nlm.nih.gov/geo) with accession number GSE133342. The source data of Figs. 1a, 1c–f, 2b–f, 3a–d, 4a–e, g, h, 5b, c, e, 6a–d, 7a–c, 8a, c, d, 9a–c, e–h, 10a, b,

d, e, Supplementary Figs. 1a–h, 2a–f, 3a–b, 4a–d, 5a–e, 6c, 7a–g, 8a–c, 9a–g, 10a–f, are provided as a Source Data file. All other data are available from the authors upon reasonable request.

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

## Acknowledgements

This work was financially supported by grants from the National Natural Science Foundation of China (31771298, 31800992, 81800512), the Natural Science Foundation of Jiangsu Province of China (BK20180554, BK20180577), the Double First-Class University Project (CPU2018GY17, CPU2018GY18), the Project of State Key Laboratory of Natural Medicines, China Pharmaceutical University (SKLNMZZRC201803), the Fundamental Research Funds for the Central Universities (2632018PY15), the Open Fund of State Key Laboratory of Pharmaceutical Biotechnology, Nanjing University, China (KF-GN-201901), the National Basic Research Program of China (973 Program) (2012CB947600, 2013CB911600), and the Priority Academic Program Development of Jiangsu Higher Education Institutions (PAPD).

## Author contributions

S.C. designed and performed research, analyzed data, and wrote the paper. M.F. performed research and analyzed data. S.Z., Z.D., Y.W., and W.Z. performed research. C.L. designed research, analyzed data, and wrote the paper.

## Additional information

**Competing interests:** The authors declare no competing interests.

