## [Peer Review File · Nature Communications]

Reviewers' comments:

Reviewer #1 (Remarks to the Author):

The main claim of the paper is that they have identified ANGPTL8 as a key regulator of the liver clock in response to food.

In their study, the authors have identified various proteins as potential regulators of the feeding and sleeping cycle through high-throughput RNA sequencing of liver samples collected from mice subjected to an overnight fasting or constant darkness. ANGPTL8 was the main target of interest as a hepatokine. They also showed that Intraperitoneal injection of recombinant Angptl8 in mice altered diurnal rhythms of locomotor activity, as well as hepatic clock and metabolic genes. This activity was dependent on transient activation of the central clock gene *Per1* and signal relay of the membrane receptor *PirB*.

I think the experiments were well designed and analysis were performed correctly.

Even though they have performed a neutralization experiment of Angptl8. This neutralization was partial. An ANGPTL8 knockout is essential to support their claim as a result this paper lacks substantial support to the claim by the authors on the role of ANGPTL8 on resetting the circadian clock in animal. Knockout animals for ANGPTL8 are available and this experiment is feasible. As a result we feel that this paper should not be accepted.

Reviewer #2 (Remarks to the Author):

Chen et al. study the role of the hepatokine Angptl8 (betatrophin) in the regulation of the liver circadian clock by food. They show that Angptl8 mRNA and peptide levels are regulated by the circadian clock and feeding in mice. Treatment with Angptl8 affects hepatocyte clock gene regulation through induction of *Per1* (and maybe *Per2*) and blocking Angptl8 release in response to food may affect feeding-mediated liver clock resetting.

Overall, this is an interesting paper introducing a new player in circadian clock entrainment. There are some experimental issues, though, that should be addressed to substantiate the findings.

Major points:

1. The authors argue that their Angptl8 treatment results in blood levels in the physiological range. Considering that Angptl8 has a half-life of 2.5h, it can be estimated that blood peak levels 11 hours after treatment would not correspond to the peak, but closer to 10-20 % of peak levels after treatment. Can one expect effects at more physiological doses?
2. Angptl8 levels are induced by feeding (which is also reflected by the levels observed after restricted feeding – Fig. 2d), but DD rhythms show peak levels at the end of the fasting phase (Fig. 1e). How can this be explained?
3. Cut-off levels for non-regulated genes in the refeed group are set at 1 which includes genes that are regulated 2fold. This seems a rather lenient cut-off.
4. At several places in the manuscript, the authors claim that Angptl8 treatment or blockage dampens or augments clock gene expression rhythms. This is not obvious from the figures (Fig. 2; 4). How was this change in amplitude assessed?
5. Did the authors consider that – at least in the mouse – Angptl8 is also produced in adipose tissues (Biochem Biophys Res Commun. 2012 Aug 10; 424(4):786-92)?
6. As a more general remark, I believe that the finding that Angptl8 may transmit time information between different hepatocytes, thus, potentially acting as an intra-tissue coupling agent outside the SCN is much more exciting than its potential role in food resetting. Unfortunately, the authors do not follow this link in the current paper.
7. Potential co-founders of the in-vivo feeding experiments such as GLP-1 (PLoS One. 2013 Nov 15;8(11):e81119), oxyntomodulin (Elife. 2015 Mar 30;4:e06253) or ghrelin are not considered.

Minor points:

1. Line 46: cite original data
2. Line 70f: does Angptl8 injection alter TG/TC rhythms?
3. Line 75f: the DD injection experiment seems flawed. Errors in assessment of CTs by monitoring activity is larger than the measured changes in period length (and this master clock effect is not reflected at the level of the SCN anyway). On the other hand, what are activity onsets on the first day in DD, when light masking is released?
4. Line 87: which statistics were used to compare expression rhythms between treatments?
5. Line 100: use „PER1“ for protein
6. Line 111: the postulated dampening of clock gene rhythms is not obvious from the figure. How was this assessed? Consider repeating this experiment in Per1-deficient mice.
7. Line 126: same as in 6
8. Line 142: the clock gene dampening is not obvious (e.g. Dbp is not dampened). One would have to monitor this over more than one cycle. Consider repeating the experiment in DD to avoid confounding effects by the light:dark cycle.
9. Fig. 2: What is clock gene expression in the SCN on the first/second day in DD (no light masking)?
10. Does Fig. 2D suggest that Per1 is down-regulated after serum shock or Angptl8 treatment? Would that contradict the data in, e.g., Fig 3?
11. Fig 4d/e: experiments should include a fasted cohort to show that refeeding actually resets the liver clock – which would then presumably be rescued by blocking Angptl8.

Reviewer #3 (Remarks to the Author):

In the current paper, the authors identified Angptl8 as a hepatokine whose levels show a daily fluctuation in the liver and serum of mice. The authors found out that injection with recombinant Angptl8 alters wheel running activity and hepatic clock gene expression in mice. Treatment with Angptl8 for 2 hours induced the rhythmic expression of clock genes in hepatoma cells. Per1 KO or knockdown of PirB, a receptor of Angptl8, abolished the actions of Angptl8. Neutralizing antibody against Angptl8 partly attenuated food-entrained resetting of liver clock in mice. The authors concluded that hepatokine Angptl8 functions as a key regulator of the liver clock in response to food intake.

Overall, this is an interesting manuscript. The authors have provided compelling evidence suggesting that hepatokine Angptl8 regulates hepatic clock gene in an autocrine/paracrine manner. This hypothesis is potentially attractive. However, it is already known that Angptl8 gene expression in the liver shows oscillatory expression in response to food intake. The overall picture of the regulatory mechanism by which Angptl8/PirB regulates Pir1 expression is unclear.

1. Please add the explanation about the known function of the molecule Angptl8 in the paragraph of the introduction. For example, Angptl8 is known to inhibit the activity of lipoprotein lipase. Of note, one early paper has already reported that fasting and feeding signals regulate Angptl8 oscillatory expression in the liver via LXRalpha and glucocorticoid receptor (Scientific Reports 2016 6 36929). Therefore, the daily fluctuation of Angptl8 hepatic expression in the current study is not a novel finding. The authors should cite this paper and should discuss the molecular mechanism by which refeeding increases gene expression of Angptl8 in the liver.
2. The authors assessed the effects of neutralizing antibody against Angptl8 in mice, but the authors should check the phenotype of Anptl8 KO mice after food intake.
3. A major flaw with this study is that the molecular mechanisms by which Angptl8-PirB axis regulates Per1 gene expression in nucleus remain unresolved. Is intracellular PirB signaling completely unknown? Does PirB signaling link to the transcriptional activity of PirB? Additional experiments are needed to address this issue.
4. In line115, the word “that” is dually used. Please remove one “that.”
5. Please add the subheadings like “introduction,” and “result” to the manuscript for general readers.

6. In the data from mice, the bar graphs only are not sufficient. Please add the individual blots of the data to the bar graphs.

Comments on the reviews

We would like to thank the reviewers for the critical evaluation of our manuscript. We think the reviewers have raised very important issues in their assessment of our study. We agree with all their comments, and have conducted a number of new studies to address these issues. As a result of these amendments, our manuscript is significantly changed and much stronger. Below, we include a point-by-point discussion of the reviewers' points, and discuss how we addressed the various issues.

Reviewer #1:

The main claim of the paper is that they have identified ANGPTL8 as a key regulator of the liver clock in response to food. In their study, the authors have identified various proteins as potential regulators of the feeding and sleeping cycle through high-throughput RNA sequencing of liver samples collected from mice subjected to an overnight fasting or constant darkness. ANGPTL8 was the main target of interest as a hepatokine. They also showed that Intraperitoneal injection of recombinant Angptl8 in mice altered diurnal rhythms of locomotor activity, as well as hepatic clock and metabolic genes. This activity was dependent on transient activation of the central clock gene *Per1* and signal relay of the membrane receptor *PirB*. I think the experiments were well designed and analysis were performed correctly. Even though they have performed a neutralization experiment of Angptl8. This neutralization was partial.

An ANGPTL8 knockout is essential to support their claim as a result this paper lacks substantial support to the claim by the authors on the role of ANGPTL8 on resetting the circadian clock in animal. Knockout animals for ANGPTL8 are available and this experiment is feasible. As a result we feel that this paper should not be accepted.

Response: We thank the reviewer for raising this important concern. We totally agree that using Angptl8 KO mice as a model will help to illustrate the role of this hepatokine in the regulation of peripheral clock. On the other hand, it does not escape from our notice that Angptl8 KO mice manifest various metabolic phenotypes, such as the significant decrease in the fat mass at the age of 12 weeks (*PNAS* 2013; 10:16109-16114). The changes in adipose tissues, a critical endocrine

organ, will lead to the alteration of circulating adipokine levels, which has a potential impact on the homeostasis of liver clock (e.g. adiponectin induces phase advance of the liver clock in KK/Ta mice). In addition, we could not exclude the possibility that the whole-body knockout of *Angptl8* may influence the central nervous system and/or cause compensatory changes in mice, and indirectly regulates the liver clock. Therefore, although *Angptl8* KO mice may serve as a feasible animal model in the present study, it is not a “clean” setting and possibly mask the direct regulation of liver clock by *Angptl8*.

As liver is the major site for *Angptl8* expression and secretion, we constructed mice with liver-specific *Angptl8* knockdown by using AAV9 system harboring an *Angptl8* shRNA. As shown in Supplementary Fig. 9a and Fig. 9a, AAV9-mediated knockdown of *Angptl8* successfully blocked the refeeding-induced *Angptl8* expression and secretion. Coincided with the findings in *Angptl8* KO mice, liver-specific *Angptl8* knockdown led to the decrease in serum TG levels of refeed mice (Fig. 9h).

To further investigate the effects of *Angptl8* on the resetting of clock and metabolic gene oscillation in mouse liver in response to fasting/refeeding cycles, we subjected these mice to DD for 7 days, and then fasted the mice for 12 h followed by a refeeding for 21 h (the experimental flowchart was shown in Fig. 9d). We found that refeeding increased the amplitude of *Per2* oscillation, whereas decreased the amplitude of *Bmal1* and *Dbp* oscillation, compared to that in the fasted cohort. In addition, refeeding signals also robustly induced phase advance of these clock gene rhythmicity. Contrasting, liver-specific *Angptl8* knockdown attenuated the amplitude of *Per2*, while restored the amplitude of *Bmal1* and *Dbp* fluctuation. At the meanwhile, *Angptl8* knockdown regulated the phase of *Bmal1* rhythmicity in the liver of refeed mice towards to that of the fasted cohort (Fig.9g and Supplementary Table 8). For the metabolic genes, the refeeding-induced pulse of *Hmgcr* oscillation was severely impaired when *Angptl8* was blocked (Supplementary Fig. 9g). These findings are consistent with the observations in mice injected with a neutralized antibody against *Angptl8* (Fig. 10d and Supplementary Fig. 10f), and confirm that *Angptl8* is a critical mediator in the food-entrained resetting of the liver clock.

Reviewer #2:

Major points:

1. The authors argue that their Angptl8 treatment results in blood levels in the physiological range. Considering that Angptl8 has a half-life of 2.5h, it can be estimated that blood peak levels 11 hours after treatment would not correspond to the peak, but closer to 10-20 % of peak levels after treatment. Can one expect effects at more physiological doses?

Response: In our experiments, we injected mice with Angptl8 at the dose of 1 µg/kg (roughly 40 ng/mouse) at ZT2, when the levels of endogenous Angptl8 reached the nadir. After injection, we examined the serum levels of Angptl8 at ZT13 and found that it was 4.52 ng/ml, which indeed fell in the range of 10-20% of exogenous Angptl8 supplement (considering the average circulating blood volume is 1.8 ml per mouse). On the other hand, in all the physiological settings we have ever checked, the peak levels of endogenous Angptl8 were around 4.88 ng/ml. Therefore, 4.52 ng/ml is also a relatively high, but still physiological dose for Angptl8's functions.

2. Angptl8 levels are induced by feeding (which is also reflected by the levels observed after restricted feeding – Fig. 2d), but DD rhythms show peak levels at the end of the fasting phase (Fig. 1e). How can this be explained?

Response: In the LD cycles, ZT9 is the end of the fasting phase. However, DD is known to shift the onset time of mice, leading to a phase advance. Therefore, after 6-week DD adaptation, CT9 can not be defined as the end of the fasting phase for mice.

3. Cut-off levels for non-regulated genes in the refed group are set at 1 which includes genes that are regulated 2 fold. This seems a rather lenient cut-off.

Response: To increase the stringency of cut-off in the refed group, we reset the cut-off levels to 0.5, thus excluding 2906 genes in the new calculation.

4. At several places in the manuscript, the authors claim that Angptl8 treatment or blockage dampens or augments clock gene expression rhythms. This is not obvious from the figures (Fig. 2; 4). How was this change in amplitude assessed?

Response: In our revision work, we calculated the phase shift and amplitude for each clock gene oscillation by fitting them to a cosine-wave equation [$y = \text{baseline} + (\text{Amplitude} \times \text{Cos}(2 \times \pi \times (x - \text{phaseshift}) / 24))$] with a fixed 24-h period (detailed data for the oscillation of clock genes were presented in Supplementary Table 8). Time series data were analyzed using one-way or two-way ANOVA followed by Bonferroni's *post hoc* test.

5. Did the authors consider that – at least in the mouse – Angptl8 is also produced in adipose tissues (Biochem Biophys Res Commun. 2012 Aug 10; 424(4):786-92)?

Response: This is a good question. We admit that in addition to the liver, adipose tissues also produce Angptl8 in mice. It is important to dissect whether Angptl8 produced either by liver or by adipose tissues can be secreted and is functional for the regulation of clocks. In our revision work, we constructed mice with liver-specific Angptl8 knockdown by using AAV9 system carrying an Angptl8 shRNA. We found that refeeding increased circulating Angptl8 levels as expected, and liver-specific knockdown of Angptl8 inhibited such an increase to that of fasting condition, indicating that liver is the major organ responsible for the secretion of Angptl8 during refeeding. Consistently, Dang et al. indicated that adenovirus-mediated (targeting the liver) overexpression of Angptl8 increased its plasma concentration, whereas a mutation ($\Delta 25$ -Angptl8) lacking the signal peptide did not secrete detectable Angptl8 in the plasma (*Scientific Report*. 2016, 6:36926). On the other hand, we examined the effects of exogenous Angptl8 injection on the mRNA expression of core clock genes in both WATs and BATs. As shown below (Fig. a), Angptl8 treatment modestly affected the oscillation of these genes in adipose tissues, indicating that the regulation of circadian clock by Angptl8 was in a liver-specific manner.

Fig. a. | Angptl8 treatment did not alter the circadian oscillations of key clock genes in WATs and BATs.

6. As a more general remark, I believe that the finding that Angptl8 may transmit time information between different hepatocytes, thus, potentially acting as an intra-tissue coupling agent outside the SCN is much more exciting than its potential role in food resetting. Unfortunately, the authors do not follow this link in the current paper.

Response: We appreciate the reviewer for raising this important point. We admit that Angptl8 may transmit time information among different liver cells, such as hepatocytes, hepatic satellite cells and Kupffer cells. Based on our findings, the local time transmission is possible because Angptl8 is predominately secreted by the liver, and regulates the liver clock in turn. Currently, the physiological roles of Angptl8 in HSC and Kupffer cells remain unknown and it is interesting to investigate this question in the future study. On the other hand, it should be noted that Angptl8 is a secreted protein and can be delivered to various peripheral tissues through circulation. This factor also responds robustly to the external stimuli, in particular nutritional signals. Therefore, we believe that evaluation of the physiological functions of Angptl8 at the organ/tissue level (when specifically considering that the resetting of peripheral clock is largely dependent on the hormone cues balanced by the organ crosstalk) gets higher priority than that at the intra-tissue level.

7. Potential co-founders of the *in vivo* feeding experiments such as GLP-1 (PLoS One. 2013 Nov 15; 8(11):e81119), oxyntomodulin (Elife. 2015 Mar 30; 4:e06253) or ghrelin are not considered.

Response: Indeed, it has been reported that various hormones, such as GLP-1, OXM, and ghrelin, play a certain role in the resetting of circadian clock. Also, these hormones can be regulated by the fasting/refeeding cycles, raising the possibility that they may be involved in the Angptl8-induced liver clock re-entrainment. In our revision work, we found that either knocked down of Angptl8 by AAV-mediated shRNA delivery, or quench of Angptl8 by the injection of a neutralized antibody, did not alter mouse serum levels of glucagon, insulin, GLP-1, OXM and ghrelin in response to the refeeding signals (Supplementary Fig. 9 and 10). These results excluded the possibility mentioned above.

Minor points:

1. Line 46: cite original data.

Response: Already cited.

2. Line 70f: does Angptl8 injection alter TG/TC rhythms?

Response: This is a good question. As shown in Fig. 2f, Angptl8 administration did not alter the rhythms of TG and TC serum levels. These results suggest that the dose of Angptl8 we selected was a physiological dose and did not cause obvious metabolic fluctuations (please also refer to the response to major point #1).

3. Line 75f: the DD injection experiment seems flawed. Errors in assessment of CTs by monitoring activity is larger than the measured changes in period length (and this master clock effect is not reflected at the level of the SCN anyway). On the other hand, what are activity onsets on the first day in DD, when light masking is released?

Response: The physiological roles of Angptl8 in the central nervous system (CNS) are still unknown. Although injection of Angptl8 in LD cycles indeed shortened the circadian period, its effects on the period were relatively minor and may be secondary due to other metabolites which

were produced in response to Angptl8 injection and were able to penetrate blood-brain barrier. That is also the reason why we did not use Angptl8 KO mice in our present study. Whole-body Angptl8 KO may cause side-effects in CNS and alter the central clock, which will in turn impact the liver clock and thus mask the direct regulation of Angptl8 in this organ. On the other hand, switch of LD to DD is actually a stress to the mice, so that the activity onsets on the first day in DD were irregular, and were thus excluded from our statistic calculations of mouse activity phase.

4. Line 87: which statistics were used to compare expression rhythms between treatments?

Response: Please refer to the response to major point #4.

5. Line 100: use “PER1” for protein.

Response: Corrected.

6. Line 111: the postulated dampening of clock gene rhythms is not obvious from the figure. How was this assessed? Consider repeating this experiment in *Per1*-deficient mice.

Response: As we showed in our original manuscript, the Angptl8 shock induced the oscillation of major clock gene expression in Hepa1c1c-7 cells, while *Per1* knockout dampened their oscillations (detailed assessment was described in response to major point #4). As a result, *Per1* deficiency alleviated the rhythmic amplitude of *Per2*, while increased the amplitude of *Bmal1* and *Dbp*. Significant phase delays of these clock genes were also observed in *Per1* KO cells (Fig. 6a and Supplementary Table 8). As the reviewer requested, we also examined the role of *Per1* in the Angptl8-induced resetting of clock gene oscillation *in vivo* by using adenovirus-mediated liver-specific *Per1* knockdown mice. We found that the amplitude of clock genes, such as *Per2* and *Dbp*, was reduced, while leaving the circadian phases unchanged (Fig. 6c and Supplementary Table 8). Taken the *in vitro* and *in vivo* observations together, we conclude that *Per1* deficiency impairs the resetting function of Angptl8 in the liver clock.

7. Line 126: same as in 6.

Response: Similar to the role of *Per1* in mediating the *Angptl8*-induced oscillation of clock genes in mouse Hepa1c1c-7 cells, knockdown of *PirB* also dampened clock gene oscillation. For example, the circadian oscillation of *Per2* was impaired (evidenced by the low R values when fitted to the cosine formula) and the amplitude of *Dbp* oscillation was increased (Fig. 8d and Supplementary Table 8). Collectively, these data suggest that *PirB* is a mediator to evoke clock gene oscillation in *Angptl8*-shocked liver cells.

8. Line 142: the clock gene dampening is not obvious (e.g. *Dbp* is not dampened). One would have to monitor this over more than one cycle. Consider repeating the experiment in DD to avoid confounding effects by the light:dark cycle.

Response: As the reviewer suggested, we performed such experiment under DD conditions to confound the effects by light:dark cycle. We first housed mice in DD for 7 days and then subjected them to a 12-h fasting followed by refeeding for 21 h. 1 h before refeeding (at CT0), an acute injection of *Angptl8*-neutralized antibody (50 μ g/25 g) or control IgG was administered to mice (the experimental flowchart was shown in Fig. 10c). We found that the neutralization of *Angptl8* antagonized the effect of refeeding on the resetting of liver clock, evidenced by the decrease in the amplitudes of *Per2* and *Dbp* mRNA oscillation and the delay in the phase of *Bmall* rhythmicity (Fig. 10d and Supplementary Table 8). Regarding the metabolic genes, the refeeding-induced pulse of *Hmgcr* oscillation was severely impaired when *Angptl8* was blocked (Supplementary Fig. 10f). The diurnal oscillation of serum levels of metabolites, especially TG, was also dampened in refed mice (Fig. 10e).

9. Fig. 2: What is clock gene expression in the SCN on the first/second day in DD (no light masking)?

Response: As shown below (Fig. b), the diurnal oscillation of key clock genes, such as *Bmall*, *Per2* and *Dbp*, was intact in the SCN of the mice kept under LD cycles. At the meanwhile, on the

first/second day, we found that majority of the clock genes still maintained similar oscillation patterns, compared to that in LD, indicating that the endogenous (or autonomous) clock is still functional in the SCN when the light cues are missing.

Fig. b. | Key clock gene expression in the SCN on the first/second day in DD. * $P < 0.05$ and ** $P < 0.01$ vs. Time 1 h.

10. Does Fig. 2d suggest that *Per1* is down-regulated after serum shock or *Angptl8* treatment? Would that contradict the data in, e.g., Fig 3?

Response: There is a misunderstanding in the interpretation of shock experiments. For the shock experiments, the cells were synchronized by *Angptl8* or 50% horse serum for 2 h and then incubated with the *Angptl8*/serum-free medium (the time was set as 0 at the replacement of medium). Therefore, at $T=0$, *Per1* was already increased by *Angptl8* (equivalent to the data in Fig. 4a in our revised manuscript). In the following observations, *Angptl8* was absent so that *Per1* acutely decreased and then began to oscillate.

11. Fig 4d/e: experiments should include a fasted cohort to show that refeeding actually resets the liver clock – which would then presumably be rescued by blocking *Angptl8*.

Response: We included a fasted cohort in our revision work. To further investigate the effects of Angptl8 on the resetting of clock and metabolic gene oscillation in mouse liver in response to fasting/refeeding cycles, we knocked down Angptl8 expression in mice by using either AAV-mediated shRNA delivery or a neutralized antibody against Angptl8 knockdown. These mice were subjected to DD for 7 days (please refer to response to minor point #8), and then were fasted for 12 h followed by refeeding for 21 h (the experimental flowchart was shown in Fig. 9d and Fig. 10c). As shown in Fig. 9g, refeeding increased the amplitude of *Per2* oscillation, whereas decreased the amplitude of *Bmal1* and *Dbp* oscillation, compared to that in the fasted cohort. In addition, refeeding signals also robustly induced phase advance of these clock gene rhythmicity. Contrasting, liver-specific Angptl8 knockdown or Angptl8 neutralization attenuated the amplitude of *Per2*, while restored the amplitude of *Bmal1* and *Dbp* fluctuation. At the meanwhile, both manipulations regulated the phase of *Bmal1* rhythmicity in the liver of refed mice towards to that of the fasted cohort (Fig. 9g and 10d, and Supplementary Table 8). Regarding the metabolic genes, the refeeding-induced pulse of *Hmgcr* oscillation was severely impaired when Angptl8 was blocked (Supplementary Fig. 9g and 10f).

Reviewer #3:

1. Please add the explanation about the known function of the molecule Angptl8 in the paragraph of the introduction. For example, Angptl8 is known to inhibit the activity of lipoprotein lipase. Of note, one early paper has already reported that fasting and feeding signals regulate Angptl8 oscillatory expression in the liver via LXR α and glucocorticoid receptor (Scientific Reports 2016, 6, 36929). Therefore, the daily fluctuation of Angptl8 hepatic expression in the current study is not a novel finding. The authors should cite this paper and should discuss the molecular mechanism by which refeeding increases gene expression of Angptl8 in the liver.

Response: We are sorry for this missing information in the Introduction section. As the reviewer suggested, we have cited this important paper and pointed out that the oscillation of Angptl8 in the liver has been reported. We also discussed the molecular mechanism by which refeeding increases the hepatic Angptl8 expression in the Discussion section as below: “At the metabolic level, recent

studies have implied an essential role of *Angptl8* in the regulation of TG homeostasis and VLDL secretion during the fasting-refeeding transition. Mechanistically, refeeding signals trigger the LXR α and RXR to form a heterodimer, and further activating the transcription of *Angptl8* through binding to the LXR response elements on its proximal promoter. In contrast, fasting signals induce the accumulation of glucocorticoids in the circulating system. This accumulation activates and recruits glucocorticoid receptor (GR) dimmers to the nGRE located on the *Angptl8* promoter region, thus inhibiting *Angptl8* transcription. Coincided with these findings, our present study indicated that the refeeding robustly increased *Angptl8* mRNA expression in 10 min (Supplementary Fig. 8a), suggesting a rapid turnover of *Angptl8* mRNA in response to the fasting and refeeding transition. It should be noted that the half-life of *Angptl8* mRNA is only 15.7 min and such an instability is important for its oscillation and function”.

2. The authors assessed the effects of neutralizing antibody against *Angptl8* in mice, but the authors should check the phenotype of *Angptl8* KO mice after food intake.

Response: We thank the reviewer for raising this important concern. We totally agree that using *Angptl8* KO mice as a model will help to illustrate the role of this hepatokine in the regulation of peripheral clock. On the other hand, it does not escape from our notice that *Angptl8* KO mice manifest various metabolic phenotypes, such as the significant decrease in the fat mass at the age of 12 weeks (*PNAS* 2013; 10:16109-16114). The changes in adipose tissues, a critical endocrine organ, will lead to the alteration of circulating adipokine levels, which has a potential impact on the homeostasis of liver clock (e.g. adiponectin induces phase advance of the liver clock in KK/Ta mice). In addition, we could not exclude the possibility that the whole-body knockout of *Angptl8* may influence the central nervous system and/or cause compensatory changes in mice, and indirectly regulates the liver clock. Therefore, although *Angptl8* KO mice may serve as a feasible animal model in the present study, it is not a “clean” setting and possibly mask the direct regulation of liver clock by *Angptl8*.

As liver is the major site for *Angptl8* expression and secretion, we constructed mice with liver-specific *Angptl8* knockdown by using AAV9 system carrying an *Angptl8* shRNA. As shown in Supplementary Fig. 9a and Fig. 9a, AAV9-mediated knockdown of *Angptl8* successfully

blocked the refeeding-induced Angptl8 expression and secretion. Coincided with the findings in Angptl8 KO mice, liver-specific Angptl8 knockdown led to the decrease in serum TG levels of refeed mice (Fig. 9h).

To further investigate the effects of Angptl8 on the resetting of clock and metabolic gene oscillation in mouse liver in response to fasting/refeeding cycles, we subjected these mice to DD for 7 days, and then fasted the mice for 12 h followed by a refeeding for 21 h (the experimental flowchart was shown in Fig. 9d). We found that refeeding increased the amplitude of *Per2* oscillation, whereas decreased the amplitude of *Bmal1* and *Dbp* oscillation, compared to that in the fasted cohort. In addition, refeeding signals also robustly induced phase advance of these clock gene rhythmicity. Contrasting, liver-specific Angptl8 knockdown attenuated the amplitude of *Per2*, while restored the amplitude of *Bmal1* and *Dbp* fluctuation. At the meanwhile, Angptl8 knockdown regulated the phase of *Bmal1* rhythmicity in the liver of refeed mice towards to that of the fasted cohort (Fig.9g and Supplementary Table 8). For the metabolic genes, the refeeding-induced pulse of *Hmgcr* oscillation was severely impaired when Angptl8 was blocked (Supplementary Fig.9g). These findings are consistent with the observations in mice injected with a neutralized antibody against Angptl8 (Fig. 10d and Supplementary Fig. 10f), and confirm that Angptl8 is a critical mediator in the food-entrained resetting of the liver clock.

3. A major flaw with this study is that the molecular mechanisms by which Angptl8-PirB axis regulates *Per1* gene expression in nucleus remain unresolved. Is intracellular PirB signaling completely unknown? Does PirB signaling link to the transcriptional activity of PirB? Additional experiments are needed to address this issue.

Response: We thank the reviewer for raising this important concern. It is indeed necessary to elucidate the molecular mechanisms by which the Angptl8-PirB axis regulates *Per1* gene expression in nucleus. Previous studies have indicated that Angptl8 increases the insulin sensitivity in HepG2 cells through triggering the phosphorylation of AKT and GSK-3 β proteins. On the other hand, PirB, acting as a receptor, responds to the external inflammatory signals or bacterial stimulation and regulates the phosphorylation of downstream MAPK and NF- κ B factors through recruitment of SH2 containing phosphatases (e.g. SHP-1/2). Notably, these factors are

intensively involved in the regulation of Per1 expression. These findings strongly suggest that various kinases (MAPK and AKT), as well as transcriptional factors (e.g. NF- κ B), may mediate the regulation of Per1 expression by the Angptl8-PirB axis.

To test our hypothesis, we first examined the effects of Angptl8 on the phosphorylation of these factors in our revision work. As shown in Fig. 5a and Supplementary Fig. 5, Angptl8 increased the phosphorylation levels of ERK1/2, P38, NF- κ B and AKT, while leaving the GSK-3 β phosphorylation unaltered. More importantly, SB203580 (P38 MAPK inhibitor), Bay11-7082 (NF- κ B inhibitor) and Wortmannin (AKT inhibitor) abrogated the Angptl8-induced Per1 transcription and translation, whereas U0126 (ERK1/2 inhibitor) failed to do that (Fig. 5b-e). These data suggest that P38 MAPK, NF- κ B, and AKT mediate the actions of Angptl8 on Per1 expression.

To get further insights into the role of PirB in the regulation of transcriptional factors by Angptl8, we specifically knocked down PirB expression in Angptl8-treated Hepa1c1c-7 cells by transducing adenoviruses expressing PirB shRNA. Our data indicated that knockdown of PirB abolished the Angptl8-induced phosphorylation of P38 MAPK, NF- κ B and AKT (Fig. 8e). Next, we treated cell with Angptl8 for 0 min, 5 min, and 10 min, respectively, and performed Co-IP analysis by using anti-PirB antibody. Our results showed that treatment of Angptl8 induced a rapid and transient reduction in PirB tyrosine phosphorylation, accompanied with a decrease in the association with SHP-1/2 (Fig. 8f).

Taken together, our new experiments demonstrated that Angptl8 induced the phosphorylation of P38 MAPK, NF- κ B and AKT through decreasing the auto-phosphorylation of PirB at tyrosine sites and reducing the recruitments of SHP1/2, and finally activated Per1 expression.

4. In line 115, the word “that” is dually used. Please remove one “that”.

Response: Corrected.

5. Please add the subheadings like “introduction,” and “result” to the manuscript for general readers.

Response: We originally submitted our manuscript in the Letter format. Now we have modified it as a full research article and all the necessary subheadings were included.

6. In the data from mice, the bar graphs only are not sufficient. Please add the individual blots of the data to the bar graphs.

Response: We have provided all the original data in the Supplementary dataset in our revised manuscript.

REVIEWERS' COMMENTS:

Reviewer #1 (Remarks to the Author):

Response is acceptable.

Reviewer #2 (Remarks to the Author):

I thank the authors for considering my various comments. The paper has much improved and, in my opinion, is now ready for publication. I have no further comments.

Reviewer #3 (Remarks to the Author):

The revision has been done very well. I have no further points.

COMMENTS ON THE REVIEW

We are delighted to see that all the reviewers are satisfied with our revision work. We think the reviewers have raised very important issues in their initial assessment of our study and therefore want to thank their great efforts with our manuscript.

REVIEWERS' COMMENTS:

Reviewer #1 (Remarks to the Author): Response is acceptable.

Response: We appreciate your contributions in reviewing our manuscript.

Reviewer #2 (Remarks to the Author): I thank the authors for considering my various comments. The paper has much improved and, in my opinion, is now ready for publication. I have no further comments.

Response: We appreciate your contributions in reviewing our manuscript.

Reviewer #3 (Remarks to the Author): The revision has been done very well. I have no further points.

Response: We appreciate your contributions in reviewing our manuscript.